# Beyond KL-Regularization: Achieving Unbiased Direct Alignment through Diffusion $f_{\chi^n}$-Preference Optimization

## Abstract

Recently, aligning diffusion models with human preferences has emerged as a key focus in text-to-image generation research. Current state-of-the-art alignment approaches predominantly rely on reverse Kullback–Leibler (KL) divergence regularization, a strategy that both restricts the potential utilization of existing data and introduces bias. In this work, we propose Diffusion-$\chi^n$PO, a novel method that refines the gradient ratio of the objective function via $f_{\chi^n}$-regularization, thereby balancing optimization power between human-preferred and non-preferred samples. Specifically, we integrate the likelihood concept of diffusion models into $\chi^2$-Preference Optimization ($\chi$PO) and re-express it as a fully differentiable objective function. Building on this foundation, we generalize to the $f_{\chi^n}$-Preference Optimization ($\chi^n$PO) framework, which substantially improves the flexibility of implicit reward model design and alleviates the influence of non-preferred samples in conflicting data. Furthermore, we provide a thorough analysis of the impacts of $\chi^2 + \text{KL}$-regularization, $f_{\chi^n}$-regularization, and KL-regularization on the alignment process from the perspective of gradient fields. Finally, we fine-tune the Stable Diffusion v1.5 model on the Pick-a-Pic preference dataset using Diffusion-$\chi^n$PO. Experimental results demonstrate enhanced alignment with textual prompts and improved visual quality, confirming the effectiveness of our proposed framework.

## 1. Introduction

Diffusion models (Croitoru et al., 2023; Ho et al., 2020; Rombach et al., 2022a) have demonstrated outstanding performance in generating realistic text-to-image synthesis;

however, a mismatch exists between their training objectives and real-world application scenarios. Due to a lack of further guidance, these models may be challenging to control effectively. Inspired by the successful application of Reinforcement Learning from Human Feedback (RLHF) in language models (Christiano et al., 2017; Rafailov et al., 2024b; Bai et al., 2022; Rafailov et al., 2024a), recent research (Fan et al., 2024; Yang et al., 2024a; Wallace et al., 2024; Liang et al., 2024; Yang et al., 2024b) has conceptualized diffusion models as a form of policy model. Under the guidance of explicit or implicit reward models learned from human-annotated preference data, the expected outputs are optimized to align generated results more closely with human preferences.

Alignment methods such as Reinforcement Learning from Human Feedback (RLHF) have achieved significant advancements in enhancing the capabilities of diffusion models. Methods such as (Clark et al., 2023; Prabhudesai et al., 2023)adjust diffusion models through pixel-level gradients derived from self-supervised reward models, while Direct Preference Optimization (DPO) (Rafailov et al., 2024b) implicitly estimates the reward model by training generative models on paired human preference data. These methods have achieved significant progress in text-to-image synthesis but remain limited by reward over-optimization. Specifically, model performance may degrade during training, as the reward model may not perfectly represent human preferences, especially in cases where the dataset does not encompass all possible scenarios. Furthermore, we observe that these methods rely on KL regularization to minimize the discrepancy between the fine-tuned model and the reference model. However, this form of regularization has been theoretically proven to be suboptimal (Zhu et al., 2023; Song et al., 2024). Additionally, the KL algorithm, when adjusting the model to reject unsafe prompts, may inadvertently introduce alignment issues. This can shift probability mass from the preferred rejection response to harmful responses, thereby reducing the likelihood of generating preferred answers and leading to over-optimization of the model.

$\chi$PO (Huang et al., 2024) improves upon the DPO algorithm by introducing a $\chi^2$-divergence term into the log-link function. Furthermore, by conducting a concentration analysis

[1]Anonymous Institution, Anonymous City, Anonymous Region, Anonymous Country. Correspondence to: Anonymous Author <anon.email@domain.com>.

Preliminary work. Under review by the International Conference on Machine Learning (ICML). Do not distribute.

focused on a single policy, it provides a theoretical guarantee of strict control over sample complexity. We extend $\chi$PO to diffusion model alignment, where the generative model is trained on paired human preference data to implicitly estimate the reward model. We derive a simple yet effective loss function for diffusion models, enabling stable and efficient preference training, which we term Diffusion-$\chi$PO. Furthermore, we propose a more general framework, Diffusion-$\chi^n$PO, which significantly enhances the design flexibility of implicit reward mode To demonstrate the effectiveness of Diffusion-$\chi^n$PO, we fine-tuned Stable Diffusion v1.5 (SD-1.5) (Rombach et al., 2022b). Experimental results indicate that models fine-tuned with Diffusion-$\chi^n$PO achieve more significant improvements across various evaluation metrics compared to those fine-tuned with existing DPO methods.

Our contributions are summarized as follows:

- We adapt the $\chi$PO framework to diffusion-based text-to-image (T2I) models by deriving a simple yet effective loss function, termed Diffusion-$\chi$PO, that enables stable and efficient preference training.

- We propose a novel $f_{\chi^n}$-regularization and present a more general framework, $\chi^n$PO, to enhance the design flexibility and specificity of implicit reward models.

- We analyzed the effects of various regularization constraints on the alignment process from the perspective of gradient ratios and validated the reliability of our method through experiments involving fine-tuned models.

## 2. Related Work

**Diffusion models** Pre-trained on internet-scale image datasets, diffusion models (Croitoru et al., 2023; Ho et al., 2020; Rombach et al., 2022a) have acquired a broad range of visual concepts and achieved significant results in text-to-image generation. However, the images generated by existing text-to-image models may still exhibit quality issues, such as inconsistencies with the input text or failure to align with human preferences.

**Alignment from human preferences** Human preferences for model outputs have been used to guide learning across a range of tasks, from behavior learning (Lee et al., 2021)to language modeling (Bai et al., 2022; Glaese et al., 2022; Ouyang et al., 2022; Liu et al., 2023; Stiennon et al., 2020), and have also been leveraged to improve the alignment of text-to-image models(Wu et al., 2023b; Lee et al., 2023). Typically, the reward model is first trained on human preference data and then fine-tuned using an online RL algorithm to maximize the scores provided by the reward model,

thereby improving the model's alignment. Compared to earlier methods that primarily focused on reward filtering or reward-weighted supervised learning, recent work has shifted towards fine-tuning policy models on feedback data (Dubois et al., 2024), or directly training policy models using a ranking loss on preference data (Rafailov et al., 2024c; Tunstall et al., 2023; Yuan et al., 2023). DPO (Rafailov et al., 2024b) proposes a supervised learning method that directly optimizes the language model from preference data, skipping reward model training and avoiding the instability of RL algorithms. Ches(Razin et al., 2024) found that during training, the likelihood of preferred responses tends to decrease in DPO. To address the bias in DPO's preference alignment process, various modifications and simplifications have been proposed, aimed at improving performance. DPO Positive (Pal et al., 2024) identified a failure mode in DPO where the standard DPO loss can decrease the likelihood of preferred responses. To address this, they propose adding a regularization term to the DPO objective to prevent this failure mode. $\chi$PO (Huang et al., 2024) modifies the logarithmic link function in the DPO objective by incorporating $\chi^2$-divergence. This addition implicitly enforces a pessimistic principle under uncertainty, thereby effectively mitigating over-optimization. Our method is inspired by DPO and $\chi$PO , and is specifically designed and adapted for diffusion models.

**Fine-tuning diffusion models on human preferences** Recently, several fine-tuning techniques have been proposed to adjust pre-trained diffusion models (Li et al., 2023; Eyring et al., 2024; Zhang et al., 2024b; Yang et al., 2024c; Deng et al., 2024a; Karthik et al., 2024; Shekhar et al., 2024; Zhang et al., 2024c), aligning them more closely with human preferences. DPOK (Fan et al., 2024) combines KL regularization with the DDPO (Black et al., 2023) loss and utilizes policy gradients to fine-tune diffusion models for achieving specific rewards. Diffusion DPO (Wallace et al., 2024) enhances the alignment of diffusion models by fine-tuning them using DPO on the Pick-a-Pic dataset, which consists of image preference pairs. D3PO (Yang et al., 2024a) proposes generating paired images from the same prompt and using either a preference model or human evaluators to identify the preferred and non-preferred images. SPO(Liang et al., 2024) improves upon DPO by incorporating a step-aware preference model and a stepwise resampling scheme. The recent DenseReward method (Yang et al., 2024b) further enhances the DPO framework by proposing a time-discounting approach that emphasizes the early denoising steps. PRDP(Deng et al., 2024b) introduces the Reward Difference Prediction (RDP) objective, which aims to enable the diffusion model to predict the reward differences between pairs of generated images. Diffusion-RPO(Gu et al., 2024) applies the RPO framework to diffusion-based text-to-image (T2I) models, simplifying the stepwise denoising

alignment loss and introducing a multimodal reweighting factor. While these studies have achieved impressive results in addressing the challenges of text-to-image alignment(Sun et al., 2024), they primarily rely on KL regularization to minimize the discrepancy between the fine-tuned model and the reference model.

## 3. Background

### 3.1. Diffusion Models

In this section, we provide a brief overview of the generative process employed by denoising diffusion probabilistic models (DDPMs). Considering a sample from the distribution $q(x_0)$ and a corresponding text prompt $c$, the text-to-image model $\pi_\theta(x_0)$, with parameters $\theta$, follows a reverse process in discrete time based on a Markov structure:

$$\pi_\theta(x_0) = \int \pi_\theta(x_{0:T}) \, dx_{1:T} = \int \prod_{t=1}^{T} \pi_\theta(x_{t-1} \mid x_t) dx_{1:T} \tag{1}$$

$x_0$ is the image, and $x_{1:T}$ represent latent variables that share the same dimensionality as $x_0$,

$$\pi_\theta(x_{t-1} \mid x_t) = \mathcal{N}\left(x_{t-1}; \mu_\theta, \sigma_t^2 \mathbf{I}\right) \tag{2}$$

is a Gaussian distribution with learnable mean and fixed covariance.

To generate an image $x_0 \sim \pi_\theta(x_0 \mid c)$, DDPM employs ancestral sampling. Given a denoising trajectory $x_{0:T}$, its log-likelihood can be analytically computed as

$$\log \pi_\theta(x_{0:T}) = \sum_{t=1}^{T} \log \pi_\theta(x_{t-1} \mid x_t)$$
$$= -\frac{1}{2} \sum_{t=1}^{T} \frac{\|x_{t-1} - \mu_\theta\|^2}{\sigma_t^2} + C \tag{3}$$

where $\mu_\theta = \frac{\sqrt{\alpha_{t-1}}}{\alpha_t}\left(x_t - \beta_t\sqrt{1 - \bar{\alpha}_t}\,\epsilon_\theta(x_t, t)\right)$

### 3.2. $\chi^2$-Preference Optimization($\chi$PO)

**Offline alignment** In the offline alignment problem, the prompt $c$ and the data pairs $x_0^+$ and $x_0^-$ come from a static dataset with human-annotated labels, where $x_0^+$ is designated as the 'winner' sample and $x_0^-$ as the 'loser' sample, they are then ranked based on the binary preference $\mathbb{P}(x_0^+ \succ x_0^- | c)$. We assume that the preferences follow the Bradley-Terry model(Bradley & Terry, 1952), which stipulates that human preferences can be expressed as:

$$p_{BT}(x_0^+ \succ x_0^- | c) = \frac{\exp(r(x_0^+, c))}{\exp(r(x_0^+, c)) + \exp(r(x_0^-, c))} \tag{4}$$

Under the Bradley-Terry model, maximum likelihood estimation is employed to learn the loss function of a reward

model parameterized by $r_\phi$ from pairwise preference data $(c, x_0^+, x_0^-)$.

$$\mathcal{L}_{\text{BT}}(\phi) = -\mathbb{E}_{c, x_0^+, x_0^-}\left[\log \sigma\left(r_\phi(c, x_0^+) - r_\phi(c, x_0^-)\right)\right] \tag{5}$$

where $\sigma$ is the sigmoid function.

**Offline RLHF with $f_\chi$-regularization.** To alleviate over-optimization, $\chi$PO (Huang et al., 2024) adopts a regularization form based on the $f_\chi$ regularization, which imposes a stricter penalty on deviations from $\pi_{\text{ref}}$ than the KL regularization. Since the $f_\chi$ regularization more effectively quantifies uncertainty compared to KL-based regularization, it helps mitigate over-optimization. By incorporating this constraint, the RL objective can be reformulated as:

$$\max_{\pi_\theta} \mathbb{E}_{\mathbf{c} \sim \mathcal{D}_{\mathbf{c}}, x_0 \sim \pi_\theta(x_0|\mathbf{c})}[r(x_0, c)] - \beta D_{f_\chi}(\pi_\theta \parallel \pi_{\text{ref}}) \tag{6}$$

Where $f_\chi(z) := \frac{1}{2}(z - 1)^2 + z \log z$, $D_{f_\chi}(\pi \parallel \pi_{\text{ref}}) = \mathbb{E}_{\mathbf{c} \sim \mathcal{D}_{\mathbf{c}}}\left[D_{\chi^2}(\pi(\cdot \mid c)\|\pi_{\text{ref}}(\cdot \mid c)) + D_{\text{KL}}(\pi(\cdot \mid c)\|\pi_{\text{ref}}(\cdot \mid c))\right]$, the hyperparameter $\beta$ controls regularization.

**$\chi$PO Objective** The link function $\phi$ in $\chi$ PO is defined as $\phi_\chi(z) := f'_\chi(z) = z + \log z$, which satisfies $0 \notin \text{dom}(f')$, and therefore, Eq. (6) in (Wang et al., 2024) is reparameterized.

$$r^*(x_0, a) = \beta\phi_\chi\left(\frac{\pi_\theta^*(x_0|c)}{\pi_{\text{ref}}(x_0|c)}\right) + const$$
$$= \beta\left[\frac{\pi_\theta(x_0^*|c)}{\pi_{\text{ref}}(x_0^*|c)} + \log\left(\frac{\pi_\theta(x_0^*|c)}{\pi_{\text{ref}}(x_0^*|c)}\right)\right] + const \tag{7}$$

As in Eq. (5), $r(c, x_0)$ is estimated using maximum likelihood training for binary classification and is expressed as follows.

$$\mathcal{L}_{\chi PO}(\phi) = -\mathbb{E}_{c, x_0^+, x_0^-}\Bigg[\log \sigma\Bigg($$
$$\phi_\chi\left(\frac{\pi_\theta(x_0^+|c)}{\pi_{\text{ref}}(x_0^+|c)}\right) - \phi_\chi\left(\frac{\pi_\theta(x_0^-|c)}{\pi_{\text{ref}}(x_0^-|c)}\right)\Bigg)\Bigg] \tag{8}$$

## 4. Method

### 4.1. $\chi$PO for Diffusion Models

A consistent dataset $D_{\text{pref}} = \{(c, x_0^+, x_0^-)\}$ is utilized, where every instance includes a prompt c and two corresponding images. Human annotations suggest that $x_0^+$ is deemed better than $x_0^-$. Our objective is to train a new model $\pi_\theta$ that is consistent with human preferences, favoring preferred generations.

The regularization in Eq.(6) cannot be computed analytically because the integral in Eq.(1) is intractable. To address this issue, we instead maximize the $f_\chi$ regularization, thereby transforming the equation into the following form:

$$\max_\theta \mathbb{E}_{c \sim D_c, x_{0:T} \sim \pi_\theta(x_{0:T}|c)}[r(c, x_{0:T})]$$
$$-\beta D_{f_\chi}[\pi_\theta(x_{0:T}|c) \| \pi_{\text{ref}}(x_{0:T}|c)] \tag{9}$$

Where $f_\chi(z) := \frac{1}{2}(z-1)^2 + z \log z$ ,which satisfies $0 \notin \text{dom}(f')$,Through derivationthe(details included in Supp B) reward function can be rewritten as:

$$r(x_0, c) = \beta \mathbb{E}_{\pi_\theta(x_{1:T}|x_0,c)} \left[\phi_\chi \left(\frac{\pi_\theta^\star(x_{0:T} \mid c)}{\pi_{\text{ref}}(x_{0:T} \mid c)}\right)\right] + const \tag{10}$$

According to Eq. (5), we define the reward loss for reverse diffusion as:

$$\mathcal{L}(\theta) = -\mathbb{E}_{(x_0^+, x_0^-) \sim D, t \sim U(0,T)} \log \sigma\left(\beta \mathbb{E}_{\substack{x_{1:T}^+ \sim \pi_\theta(x_{1:T}^+|x_0^+), \\ x_{1:T}^- \sim \pi_\theta(x_{1:T}^-|x_0^-)}} \right.$$
$$\left. \left[\phi_\chi \left(\frac{\pi_\theta^\star(x_{0:T}^+ \mid c)}{\pi_{\text{ref}}(x_{0:T}^+ \mid c)}\right) - \phi_\chi \left(\frac{\pi_\theta^\star(x_{0:T}^- \mid c)}{\pi_{\text{ref}}(x_{0:T}^- \mid c)}\right)\right]\right) \tag{11}$$

Starting from Equation Eq. (11), we substitute the reverse decompositions for $\pi_\theta$ and $\pi_{\text{ref}}$ , and apply Jensen's inequality alongside the convexity of the function $-\log \sigma$ to move the expectation outward(details included in Supp C). After simplification, we derive the following bound:

$$\mathcal{L}(\theta) \leq -\mathbb{E}_{\substack{(x_0^+, x_0^-) \sim D, x_{t-1,t}^+ \sim p_\theta(x_{t-1,t}^+|x_0^+) \\ t \sim U(0,T), x_{t-1,t}^- \sim p_\theta(x_{l,t-1,t}^-|x_0^-)}} \log \sigma\left(\beta T \left[ \right.\right.$$
$$\left.\left. \phi_\chi \left(\frac{\pi_\theta^\star(x_{t-1}^+ \mid x_t^+, c)}{\pi_{\text{ref}}(x_{t-1}^+ \mid x_t^+, c)}\right) - \phi_\chi \left(\frac{\pi_\theta^\star(x_{t-1}^- \mid x_t^-, c)}{\pi_{\text{ref}}(x_{t-1}^- \mid x_t^-, c)}\right)\right]\right) \tag{12}$$

Be aware that $\pi_\theta(x_t \mid x_{t+1})$ is defined using the same formula for both preferred and rejected sample pairs. By substituting Eq.(2) into Eq.(12) (a detailed derivation is provided in Supp. D), we obtain the final Diffusion-$\chi$PO loss function.

$$\mathcal{L}(\theta) = -\mathbb{E}_{\substack{(x_0^+, x_0^-) \sim D, t \sim U(0,T), \\ x_t^+ \sim q(x_t^+|x_0^+), x_t^- \sim q(x_t^-|x_0^-)}} \log \sigma($$
$$-\beta T \left[\phi_\chi \left(\exp \Delta\epsilon \left(x_t^+, t\right)\right) - \phi_\chi \left(\exp \Delta\epsilon \left(x_t^-, t\right)\right)\right]) \tag{13}$$

where $\Delta\epsilon(x_t^*, t) = \omega(\|\epsilon_\theta(x_t^*, t) - \epsilon_t^*\|_2^2 - \|\epsilon_{\text{ref}}(x_t^*, t) - \epsilon_t^*\|_2^2)$ and $\omega = \frac{\beta_t \alpha_{t-1}}{2(1-\bar{\alpha}_{t-1})\alpha_t}$ (constant in practice (Ho et al., 2020; Kingma et al., 2021)).

---

**Algorithm 1** $f_{\chi^n}$-Preference Optimization($\chi^n$PO)

**Require:** Reference policy $\pi_{\text{ref}}$, preference dataset $\mathcal{D}_{\text{pref}}$, $f_{\chi^n}$-regularization coefficient $\beta > 0$.
1: Define:
$$\phi_{\chi^n}(z) := \frac{1}{n}(\sum_{k=1}^n \frac{1}{k} z^k + \log z)$$

2: Optimize $f_{\chi^n}$-regularized preference optimization objective:

$$\mathcal{L}_{\chi^n PO}(\theta) = -\mathbb{E}_{c, x_0^+, x_0^-} \left[\log \sigma\left(\right.\right.$$
$$\left.\left. \beta\phi_{\chi^n} \left(\frac{\pi_\theta(x_0^+|c)}{\pi_{\text{ref}}(x_0^+|c)}\right) - \beta\phi_{\chi^n} \left(\frac{\pi_\theta(x_0^-|c)}{\pi_{\text{ref}}(x_0^-|c)}\right)\right)\right] \tag{18}$$

Update model parameters $\theta$ by gradient descent

---

### 4.2. $f_{\chi^n}$-Preference Optimization($\chi^n$PO)

Our primary Algorithm 1, denoted as $\chi^n$PO , updates the policy parameters $\theta$ by solving the optimization objective defined in Eq.(18). The objective replaces the term $\phi_{\chi^n}(\frac{\pi_\theta(x_0|c)}{\pi_{\text{ref}}(x_0|c)}) := \frac{\pi_\theta(x_0|c)}{\pi_{\text{ref}}(x_0|c)} + \log \frac{\pi_\theta(x_0|c)}{\pi_{\text{ref}}(x_0|c)}$ in the original $\chi$ PO target (Eq. (8) ) with a novel link function, which is defined as follows:

$$\phi_{\chi^n}(\frac{\pi_\theta(x_0 \mid c)}{\pi_{\text{ref}}(x_0 \mid c)}) := \frac{1}{n}\left[\sum_{k=1}^n \frac{1}{k} \left(\frac{\pi_\theta(x_0 \mid c)}{\pi_{\text{ref}}(x_0 \mid c)}\right)^k \right.$$
$$\left. + \log \frac{\pi_\theta(x_0 \mid c)}{\pi_{\text{ref}}(x_0 \mid c)}\right] \tag{14}$$

**Algorithm derivation** We begin with the regularized function $f_\chi(z) := \frac{1}{2}(z-1)^2 + z \log z$. To further develop this function, we incorporate higher-order polynomial terms:

$$f_{\chi^n}(z) := \frac{1}{n} \left[\sum_{k=2}^n \frac{1}{k(k+1)} z^{k+1} + \frac{1}{2}(z-1)^2 + z \log z\right] \tag{15}$$

where $n \geq 2$ .Thus, Eq. (9) can be reformulated as:

$$\mathbb{E}_{\mathbf{c} \sim \mathcal{D}_{pref}, x_0 \sim p_\theta(x_0|\mathbf{c})}[r(x, c)]$$
$$-D_{f_{\chi^n}}[\pi_\theta(x_0|c) \| \pi_{\text{ref}}(x_0|c)] \tag{16}$$

The link function $\phi_{\chi^n}$ in $\chi^n$PO is defined as $\phi_{\chi^n}(z) = f'_{\chi^n}(z) = \frac{1}{n}(\sum_{k=1}^n \frac{1}{k} z^k + \log z)$, which satisfies $0 \notin \text{dom}(f')$ , and therefore Eq (16) in (Wang et al., 2024) is reparameterized

$$r(x_0, c) = \beta\phi_{\chi^n}(\frac{\pi_\theta^\star(x_0 \mid c)}{\pi_{\text{ref}}(x_0 \mid c)}) + const \tag{17}$$

As shown in Eq.(5), the loss function for $r(c, x_0)$ is expressed in Eq.(18)

**Diffusion-$\chi^n$PO Objective** Following the derivation process in Sec.4.1 , the final Diffusion-$\chi^n$PO loss function is

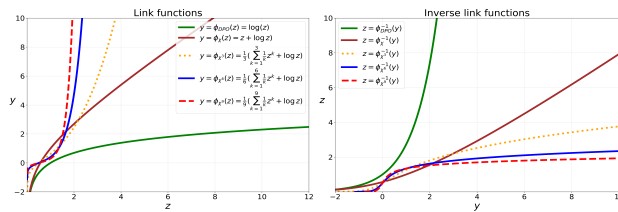

Figure 1. We compared the behavior of the $f_{\chi^n}$-regularization link function $\phi_{\chi^n}(z) = \frac{1}{n}(\sum_{k=1}^n \frac{1}{k}z^k + \log z)$ and its inverse $\phi_{\chi^n}^{-1}(z)$ with those of the Kullback-Leibler (KL) regularization link function $\phi_{\text{DPO}}(z) = log(z)$ and its inverse $\phi_{\text{DPO}}^{-1}(z) = exp(z)$.

expressed as:

$$\mathcal{L}(\theta) = -\mathbb{E}_{(x_0^+, x_0^-) \sim D, t \sim U(0,T), x_t^+ \sim q(x_t^+|x_0^+), x_t^- \sim q(x_t^-|x_0^-)}$$

$$\log \sigma(-\beta T[\phi_{\chi^n}(\exp \Delta\epsilon(x_t^+, t)) - \phi_{\chi^n}(\exp \Delta\epsilon(x_t^-, t))]) \tag{19}$$

Here, $\Delta\epsilon(x_t^*, t)$ is defined as $\omega(\|\epsilon_\theta(x_t^*, t) - \epsilon_t^*\|_2^2 - \|\epsilon_{\text{ref}}(x_t^*, t) - \epsilon_t^*\|_2^2)$.

### 4.3. Analysis on Gradient Fields of Alignment Process

To explore the characteristics of the $\chi^n$PO algorithm, we rewrite the link function formula Eq. (7) into the following form:

$$Z = \frac{\pi_\theta^*(x_0|c)}{\pi_{\text{ref}}(x_0|c)} = \phi_{\chi^n}^{-1}((r^*(x_0, a) - const)/\beta) \tag{20}$$

Here, $\phi_{\chi^n}^{-1}(z)$ represents the inverse function of the function $\phi_{\chi^n}(z)$. In Fig. 1, the inverse link function $\phi^{-1}(z)$ for $f_{\chi^n}$-regularization closely satisfies $\phi_{\chi^n}^{-1}(z) \approx z^{\frac{1}{n}}$ for $z \geq 1$ and $\phi_{\chi^n}^{-1}(z) \approx e^{z/n}$ for $z \leq 1$, while an increase in the parameter $n$ gradually decreases the slope of $\phi^{-1}(z)$, producing a "flattening" effect. Consequently, During training, this flattening can help mitigate over-optimization by reducing the amplification of extreme z values, thereby preventing excessively aggressive parameter updates or selections in optimization-based methods.

Furthermore, by abstracting away the specific characteristics of the link function $\phi(\cdot)$ and focusing on the general form of the loss function in Eq. (8), we obtain:

$$\mathcal{L}_\phi(Z_1, Z_2) = -\mathbb{E}\left[\log \sigma\left(\beta\left(\phi(Z_1) - \phi(Z_2)\right)\right)\right] \tag{21}$$

Where, $Z_1$ is defined as the training win ratio $\frac{p_\theta(x_{0:T}^w|c)}{p_{\text{ref}}(x_{0:T}^w|c)}$, and $Z_2$ corresponds to the training loss ratio $\frac{p_\theta(x_{0:T}^l|c)}{p_{\text{ref}}(x_{0:T}^l|c)}$. Thus, the expression for the gradient ratio of $\mathcal{L}_\phi(Z_1, Z_2)$ when enhancing the probability of human-preferred responses ( $Z_1$ ) versus reducing the probability of human-dispreferred responses ( $Z_2$ ) is given by:

$$\left|\frac{\partial \mathcal{L}_\phi(Z_1, Z_2)}{\partial Z_1} \middle/ \frac{\partial \mathcal{L}_\phi(Z_1, Z_2)}{\partial Z_2}\right| = \frac{\phi'(Z_1)}{\phi'(Z_2)} \tag{22}$$

Table 1. **Several link functions and their derivatives.**

|  | $\phi(z)$ | $\phi'(z)$ |
|---|---|---|
| DPO | $\log z$ | $\frac{1}{z}$ |
| $\chi$PO | $z + \log z$ | $1 + \frac{1}{z}$ |
| $\chi^n$PO | $\frac{1}{n}(\sum_{k=1}^n \frac{1}{k}z^k + \log z)$ | $\frac{1}{n}(\sum_{k=0}^n z^{k-1})$ |

According to Table 1,different regularization link function result in distinct gradient ratios. if the regularization link function measure is $\phi_{DPO}$, then the gradient ratio becomes $\frac{Z_2}{Z_1}$; if it is the $\phi_\chi$, the gradient ratio is given by $\frac{Z_2(Z_1+1)}{Z_1(Z_2+1)}$ .if it is the $\phi_{\chi^n}$, the gradient ratio is given by $\frac{\sum_{k=0}^n Z_1^{k-1}}{\sum_{k=0}^n Z_2^{k-1}}$ (details included in Supp E ).

Furthermore, as the alignment progresses, the value of $Z_1$ tends to exceed unity, while $Z_2$ tends to fall below unity. Consequently, for any pairwise preference data, the following inequality holds:

$$\frac{Z_2}{Z_1} < \frac{Z_2(Z_1+1)}{Z_1(Z_2+1)} < \frac{\sum_{k=1}^n Z_1^{k-2}}{\sum_{k=1}^n Z_2^{k-2}} < \frac{\sum_{k=1}^{n+1} Z_1^{k-2}}{\sum_{k=1}^{n+1} Z_2^{k-2}} \tag{23}$$

This inequality remains valid throughout the alignment process.The gradient ratio of DPO is smaller than that of $\chi$PO and is less than 1. When the gradient ratio falls below 1, a smaller ratio causes the probabilities of less preferred images to decrease faster than those of preferred images. This rapid decrease can inadvertently lead to misalignment, shifting probability mass from desired rejection responses to harmful responses. In contrast, the gradient of $\chi$PO is closer to 1, enabling reinforcement learning to strike a balance between reward maximization and constraint satisfaction. This property effectively alleviates over-optimization and misalignment issues, while significantly enhancing training stability and optimization efficiency. As n increases, the gradient ratio of $\chi^n$PO not only grows progressively but also exceeds 1, encouraging fine-tuned diffusion models to prioritize the optimization of human-preferred images while reducing the penalization of less preferred behaviors. This mechanism effectively alleviates inherent conflicts within human preference data pairs, significantly enhancing the efficiency of preference objective optimization and accelerating the model training process.

## 5. Experiments

Detailed implementation and evaluation procedures, along with our ablation results, are presented. We perform an extensive quantitative and qualitative evaluation of Diffusion-$\chi^n$PO to demonstrate the efficacy of the proposed $f_{\chi^n}$ regularization in fine-tuning text-to-image diffusion models for matching preference distributions.The appendix includes a

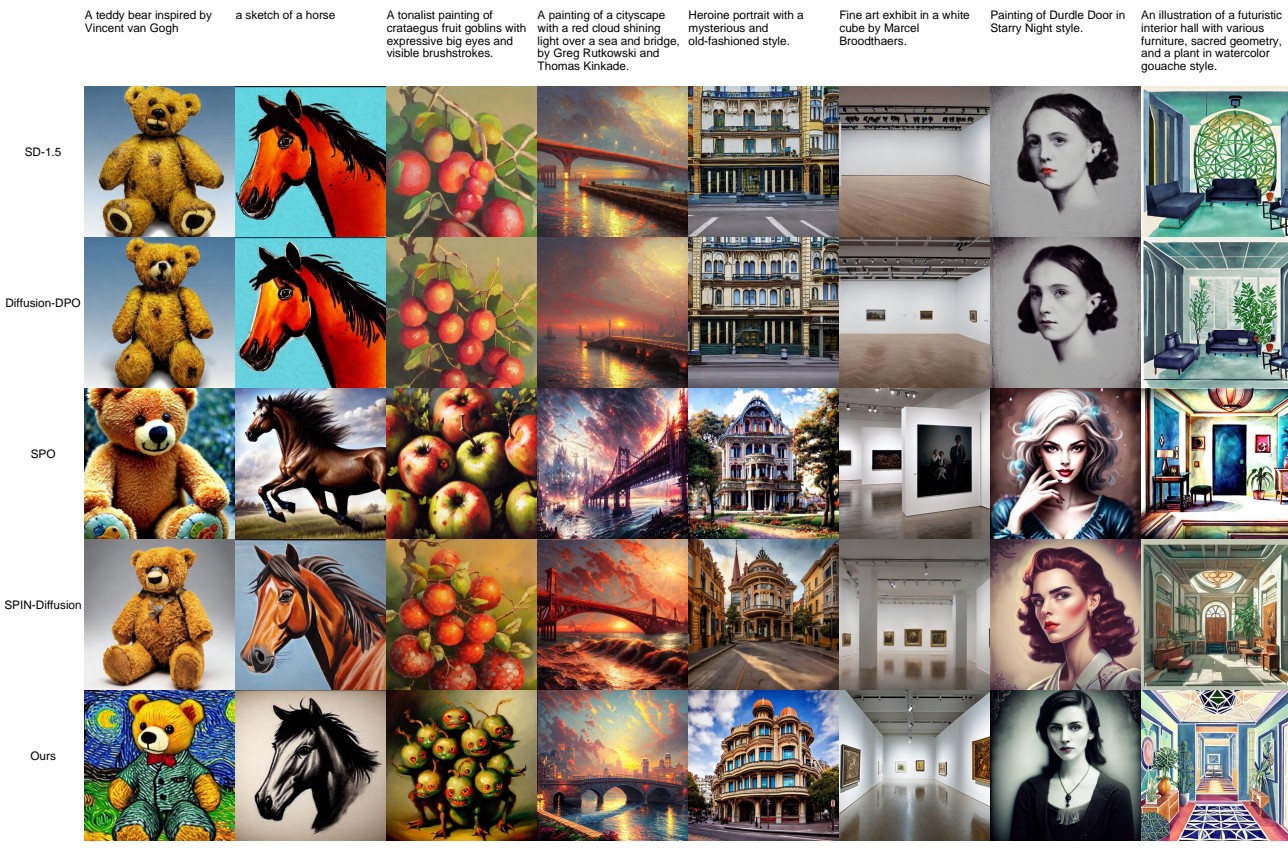

*Figure 2.* Generated images from our method and baseline models, along with their corresponding prompts from the multiple-prompt experiment, are presented here, with additional examples provided in Appendix G.

detailed comparative of our method against baseline models. Specifically, Appendix A presents reward model evaluation scores based on the HPDv2 and Parti datasets, while Appendix G showcases images generated from prompts. These results demonstrate that Diffusion $\chi^n$PO holds promise for fine-tuning text-to-image diffusion models to accommodate specific user preferences.The code for this study will be made publicly available, and the pseudocode for the training objectives is presented in Appendix F

### 5.1. Implementation details

We developed Diffusion-$\chi^n$PO by building upon the Diffusion-DPO codebase, adhering to the methodologies established in Diffusion-DPO research. For all experiments, we utilized Stable Diffusion (SD) v1.5 (Rombach et al., 2022b) as the pretrained diffusion model and fine-tuned the complete UNet weights. Large-scale fine-tuning was performed on the training set of the Pick-a-Pic v2 dataset (Kirstain et al., 2023)(MIT license). using Eq (19), demonstratingDiffusion-$\chi^n$PO's superior generation quality on complex and previously unseen prompts. We maintained the parameter settings from Diffusion-DPO, conducting

training on two NVIDIA 4090 GPUs with a local batch size of one pair and gradient accumulation over 128 steps. The training protocol included a $25\%$ linear warmup phase, a learning rate of $1 \times 10^{-8}$, and fine-tuning SD1.5 with $\beta$ set to 1000.

### 5.2. Experiment protocol

**Evaluation dataset** Following (Wallace et al., 2024), we adopt four benchmark categories from HPDv2 (Wu et al., 2023a)—animation, concept art, painting, and photography—with each category consisting of 800 prompts. In addition, 1,632 prompts from the PartiPrompts dataset (Yu et al., 2022b)are included in the evaluation dataset.

**Baselines** To evaluate the effectiveness of our proposed method, we compare it with several state-of-the-art techniques for human preference learning. The baseline methods include Diffusion-DPO (Wallace et al., 2024), Stable Diffusion 1.5 (Rombach et al., 2022b), SePPO (Zhang et al., 2024a), SPIN-Diffusion (Chen et al., 2024), Diffusion-KTO (Li et al., 2024), and SPO (Liang et al., 2024). We reproduce each of these baselines using the official check-

*Table 2. $\chi^n$PO Reward score comparison training.* Using the HPSV2 datasets, reward model scores are computed to assess Diffusion-$\chi^n$PO and Diffusion-$\chi^n$PO. As $n$ increases in the $\chi^n$PO framework, both speed and quality improve, with the highest score achieved at n = 6 . However, further increases in n result in a decline in performance. This can be attributed to the inherent conflicts in image quality within the Pick-a-Pic V2 dataset, as well as the specific characteristics of the $\chi^n$PO algorithm. A detailed discussion of these aspects can be found in Section 5.3

| Method | HPSV2 ↑ | Pick ↑ | Aesth↑ | CLIP ↑ | ImaR ↑ |
|---|---|---|---|---|---|
| SD v1-5 | 26.97 | 20.69 | 5.46 | 0.349 | 0.125 |
| Diffusion-DPO | 27.28 | 21.12 | 5.56 | 0.354 | 0.315 |
| Diffusion-$\chi$PO | 27.83 | 21.53 | 5.64 | 0.357 | 0.643 |
| Diffusion-$\chi^3$PO | 27.91 | 21.63 | 5.69 | 0.356 | 0.678 |
| Diffusion-$\chi^5$PO | 27.92 | 21.60 | 5.66 | 0.356 | 0.711 |
| Diffusion-$\chi^6$PO | **27.98** | 21.68 | 5.68 | **0.357** | **0.730** |
| Diffusion-$\chi^7$PO | 27.95 | 21.70 | 5.70 | 0.355 | 0.713 |
| Diffusion-$\chi^9$PO | 27.93 | **21.75** | 5.70 | **0.357** | 0.698 |

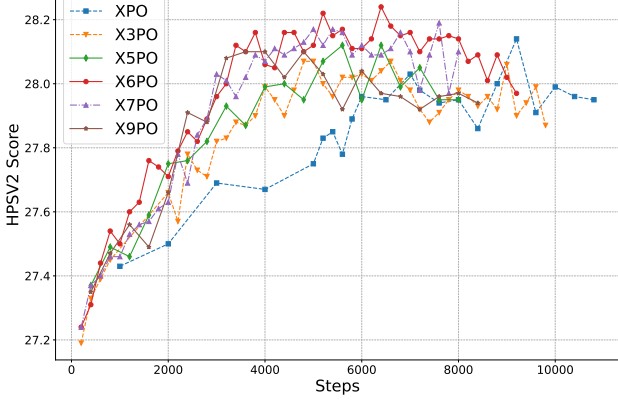

*Figure 3.* The checkpoint was evaluated on the first 100 prompts in the anime style from the HPv2 dataset, and the reward model scores for HPSv2 were computed accordingly

points available on HuggingFace.

**Metrics** We use widely recognized metrics to evaluate our approach, including PickScore (Kirstain et al., 2023) (general human preference), HPSV2 (Wu et al., 2023a) (prompt alignment), ImageReward (Xu et al., 2024) (general human preference), Aesthetic (Schuhmann, 2022) (visual appeal), and CLIP (Radford et al., 2021) (image-text alignment performance). Table 4 reports the average scores between Diffusion-$\chi^n$PO and the baselines, computed from HPDv2 and Parti (Yu et al., 2022a) validation prompts.

### 5.3. Effect of $f_{\chi^n}$ Regularization

During the training of Diffusion-$\chi^n$PO on a small dataset, the loss value initially declined sharply but then rapidly rebounded, indicating premature overfitting. This pattern led to repetitive images with pronounced noise artifacts. In contrast, when employing the Pick-a-Pic-V2 dataset—which contains over 800,000 images—this trend was absent. The dataset's breadth and diversity likely prevented early overfitting and enabled a stable reduction in loss, culminating in improved image quality over the course of training.

To investigate the impact of the hyperparameter n in $\chi^n$PO, we held all other factors constant and tested $n \in \{1, 3, 5, 6, 7, 9\}$ in comparison to a baseline model. The reward scores, evaluated on the HPDv2 validation set, are presented in Table 2, and the evolution of these scores throughout training is depicted in Fig. 3. Although the optimization efficiency of Diffusion-$\chi$PO remains below expectations, Diffusion-$\chi^n$PO achieves enhanced image quality within fewer training steps as n increases.

We hypothesize that these findings can be attributed to the inherent conflicts within certain preference pairs in the Pick-a-Pic-V2 dataset, which impede the balanced opti-

mization strategy of $\chi$PO. When the gradient ratio exceeds 1, the penalty for non-preferred behaviors is slightly reduced, thereby mitigating interference from non-preferred samples in conflicting pairs. Consequently, the optimization trajectory becomes smoother, leading to notable performance improvements. Among the configurations tested, Diffusion-$\chi^6$PO offers an optimal balance between prioritizing preferred samples and curtailing the detrimental impact of conflicting data. However, further increasing n to 9 (as in Diffusion-$\chi^9$PO) disproportionately weakens the penalty on negative samples, diminishing constraints on non-preferred behaviors and ultimately degrading overall image quality.

### 5.4. Main Results

**Quantitative results** Table 3 presents the win rates of Diffusion-$\chi^6$PO-aligned SD v1-5 relative to various baseline models across multiple automated metrics. The results demonstrate that Diffusion-$\chi^6$PO significantly improves the alignment performance of SD v1-5, outperforming existing methods in most evaluation metrics. Specifically, Diffusion-$\chi^6$PO surpasses Diffusion-DPO, SPO, SePPO, and SPIN-Diffusion on the HPSV2, PickScore, CLIP, and Image Reward metrics. Notably, it exceeds the performance of SPO by up to 60% on CLIP, Image Reward, and PickScore. In addition, Diffusion-$\chi^6$PO outperforms Diffusion-KTO on three of the five metrics, with only a slight shortfall on HPS and Aesthetic. Compared to Diffusion-DPO, these consistent improvements underscore the effectiveness of applying $f_{\chi^n}$ regularization to align SD v1.5. Moreover, Diffusion-$\chi^n$PO enables more efficient utilization of the training data, strengthens image preference learning, and boosts the model's performance across multiple evaluation metrics.

*Table 3.* **Automatic win rates** (%) of Diffusion-$\chi^6$PO (SD v1-5) compared to existing alignment approaches, utilizing prompts from the HPDv2 and Parti sets. Generated outputs were evaluated using reward models that assigned scores to each method. The method with the higher score received 1 point, while ties resulted in both methods receiving 0.5 points each. Bold is used to indicate the win rates that exceed 50%.

| Dataset | Method | HPSV2↑ | PickScore↑ | Aesthetic↑ | CLIP↑ | Image Reward↑ |
|---|---|---|---|---|---|---|
| HPSV2 | vs. SD v1-5 | **84.31** | **84.59** | **69.69** | **55.66** | **76.00** |
| | vs. Diffusion-DPO (Wallace et al., 2024) | **77.72** | **73.88** | **60.62** | **50.28** | **69.22** |
| | vs. SPO (Liang et al., 2024) | **63.80** | **56.13** | 44.63 | **75.41** | **63.09** |
| | vs. Diffusion-KTO (Li et al., 2024) | **50.11** | **67.91** | 47.09 | **54.63** | **50.78** |
| | vs. SePPO (Zhang et al., 2024a) | **53.97** | **58.16** | 40.94 | **51.97** | **53.47** |
| | vs. SPIN-Diffusion (Chen et al., 2024) | **59.62** | **54.03** | 30.69 | **61.72** | **57.78** |
| PartiPrompts | vs. SD v1-5 | **75.29** | **75.87** | **69.26** | **56.28** | **68.65** |
| | vs. Diffusion-DPO | **69.17** | **66.07** | **60.56** | **51.87** | **63.63** |
| | vs. SPO | **61.02** | **58.79** | 42.80 | **70.24** | **60.26** |
| | vs. Diffusion-KTO | 45.28 | **64.30** | 49.05 | **57.50** | **53.52** |
| | vs. SePPO | **51.53** | **55.05** | 42.87 | **54.44** | **54.75** |
| | vs. SPIN-Diffusion | **56.92** | **52.73** | 33.80 | **63.63** | **60.32** |

**Qualitative results** Figure 2 compares the images generated by Diffusion-$\chi^6$PO, SD-1.5, SPO, SPIN-Diffusion, and SePPO. As shown, Diffusion-$\chi^n$PO generally achieves better text-image alignment, producing more vivid images from both simple prompts and more challenging long-text prompts, often surpassing the baselines. Specifically, in the first column, most models fail to generate "a teddy bear inspired by Vincent van Gogh" as instructed. By contrast, Diffusion-$\chi^n$PO accurately captures the essential elements (i.e., Vincent van Gogh) and produces higher-quality results compared to Diffusion-DPO. In the second column, while most models struggle to create images in the specified sketch style, Diffusion-$\chi^n$PO successfully incorporates this key characteristic. Overall, Figure 2 provides further evidence that Diffusion-$\chi^n$PO not only enhances text-image alignment but also significantly improves the visual quality of the generated images.

## 6. Conclusion

In this paper, we propose Diffusion-$\chi^n$PO, an extended alignment framework for text-to-image models. Our method leverages $f_{\chi^n}$-regularization to enhance uncertainty quantification and mitigate the risk of over-optimization. Experimental results on the Stable Diffusion 1.5 (SD-1.5) model demonstrate that Diffusion-$\chi^n$PO achieves superior post-fine-tuning performance compared to state-of-the-art approaches, underscoring its efficacy as a robust alignment strategy in text-to-image synthesis pipelines.

## Limitations

Diffusion-$\chi^n$PO substantially enhances the alignment performance of text-to-image (T2I) diffusion models, yet cer-

tain limitations remain. Analyzing the gradient ratios for different values of n indicates that $\chi$PO yields a ratio closest to 1, which is theoretically optimal. Nonetheless, empirical results suggest that Diffusion-$\chi^6$PO achieves superior alignment outcomes in practice. A potential explanation lies in the nature of the Pick-a-Pic V2 preference dataset used during training, which comprises user-submitted prompts alongside images generated by various existing T2I models. This dataset inevitably introduces inconsistencies: some negative samples may align well with the text despite being labeled negatively, whereas some positive samples may favor inappropriate images. As a result, striking a suitable balance between positive and negative instances becomes more challenging. We further hypothesize that the inherent characteristics of various dataset may influence the optimal parameter n for the $\chi^n$PO objective, warranting a deeper investigation in future research.

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

## A. Diffusion-$\chi^n$PO resault

Table 4 summarizes the evaluation results for HPSv2 (Apache-2.0 license) and PartiPrompts (Apache-2.0 license). The findings reveal that Diffusion-$\chi^n$PO achieves state-of-the-art performance on several metrics for a wide range of prompts, and remains on par with the current leading approaches for the other metrics.

*Table 4.* **Quantitative Evaluation on HPSv2 and PartiPrompt Datasets** We conducted a comprehensive evaluation of model performance on the HPSv2 and PartiPrompt datasets, reporting various quantitative metrics. For each metric, the method achieving the highest average score is highlighted in bold.

| Test Dataset | Method | HPSV2↑ | PickScore↑ | Aesthetic↑ | CLIP↑ | Image Reward↑ |
|---|---|---|---|---|---|---|
| HPSV2 | SD v1-5 | 26.97 | 20.69 | 5.46 | 0.349 | 0.125 |
| | Diffusion-DPO | 27.28 | 21.12 | 5.56 | 0.354 | 0.315 |
| | Diffusion-$\chi$PO(ours) | 27.83 | 21.53 | 5.64 | 0.357 | 0.643 |
| | Diffusion-$\chi^3$PO(ours) | 27.91 | 21.63 | 5.69 | 0.356 | 0.678 |
| | Diffusion-$\chi^5$PO(ours) | 27.92 | 21.60 | 5.66 | 0.356 | 0.711 |
| | Diffusion-$\chi^6$PO(ours) | **27.98** | 21.68 | 5.68 | **0.357** | **0.730** |
| | Diffusion-$\chi^7$PO(ours) | 27.95 | 21.70 | 5.70 | 0.355 | 0.713 |
| | Diffusion-$\chi^9$PO(ours) | 27.93 | **21.75** | 5.70 | **0.357** | 0.698 |
| | SPO | 27.64 | 21.50 | 5.75 | 0.318 | 0.320 |
| | Diffusion-KTO | **27.99** | 21.32 | 5.70 | 0.352 | 0.689 |
| | SPIN-Diffusion | 27.76 | 21.56 | **5.89** | 0.341 | 0.543 |
| | SePPO | 27.88 | 21.50 | 5.76 | 0.354 | 0.616 |
| PartiPrompts | SD v1-5 | 26.96 | 21.24 | 5.26 | 0.34 | 0.40 |
| | Diffusion-DPO | 27.19 | 21.49 | 5.34 | 0.34 | 0.40 |
| | Diffusion-$\chi$PO(ours) | 27.54 | 21.75 | 5.41 | **0.35** | 0.64 |
| | Diffusion-$\chi^3$PO(ours) | 27.59 | 21.79 | 5.45 | 0.34 | 0.70 |
| | Diffusion-$\chi^5$PO(ours) | 27.54 | 21.75 | 5.43 | **0.35** | 0.70 |
| | Diffusion-$\chi^6$PO(ours) | 27.66 | 21.82 | 5.44 | **0.35** | **0.73** |
| | Diffusion-$\chi^7$PO(ours) | 27.63 | 21.83 | 5.44 | **0.35** | 0.70 |
| | Diffusion-$\chi^9$PO(ours) | 27.61 | **21.84** | 5.43 | **0.35** | 0.70 |
| | SPO | 27.35 | 21.57 | 5.52 | 0.32 | 0.40 |
| | Diffusion-KTO | **27.74** | 21.54 | 5.47 | 0.34 | 0.63 |
| | SPIN-Diffusion | 27.47 | 21.70 | **5.63** | 0.32 | 0.43 |
| | SePPO | 27.61 | 21.67 | 5.50 | **0.35** | 0.57 |

*Table 5.* **Per-style metric scores of Diffusion-$\chi^6$PO on the HPSv2 test set.**

| Style | HPS↑ | Pick Score↑ | Aesthetic↑ | Clip↑ | Image Reward↑ |
|---|---|---|---|---|---|
| anime | 28.48 | 21.920 | 5.54 | 0.363 | 0.838 |
| paintings | 27.88 | 21.459 | 6.01 | 0.365 | 0.792 |
| concept-art | 27.76 | 21.359 | 5.81 | 0.357 | 0.826 |
| photo | 27.80 | 21.973 | 5.35 | 0.343 | 0.465 |
| Average | 27.98 | 21.678 | 5.68 | 0.357 | 0.730 |

## B. Maximizer of the Lower Bound of RLHF Objective

**Lemma 1** Define

If $\pi_{\text{ref}}(x_{0:T} \mid c) > 0$ holds for all condition c, $f'(z)$ is an invertible function and 0 is not in definition domain of function $f'(z)$, the reward class consistent with Bradley-Terrry model can be reparameterized with the policy preference $\pi_\theta(x_{0:T})$ and the reference preference $\pi_{\text{ref}}(x_{0:T} \mid c)$ as: Assuming $\pi_{\text{ref}}(x_{0:T} \mid c) > 0$ for all conditions c, that $f'(z)$ is invertible, and that 0 is outside the domain of $f'(z)$, we can reparameterize the Bradley-Terry-based reward class in terms of the policy preference $\pi_\theta(x_{0:T})$ and the reference preference $\pi_{\text{ref}}(x_{0:T} \mid c)$ as follows:

$$r^\star(x_{0:T}, c) = \beta\phi\left(\frac{\pi_\theta^\star(x_{0:T}|c)}{\pi_{\text{ref}}(x_{0:T}|c)}\right) + const \tag{24}$$

Proof. Consider the following optimization problem:

$$\min_{\pi_\theta} -\mathbb{E}_\pi[r(c, x_{0:T})] + \beta D_f(\pi_\theta(x_{0:T} \mid c) \parallel \pi_{\text{ref}}(x_{0:T} \mid c)) \tag{25}$$

$$\text{s.t.} \sum_{x_{0:T}} \pi_\theta(x_{0:T} \mid c) = 1, \pi_\theta(x_{0:T} \mid c) \geq 0 \quad \forall x_{0:T}. \tag{26}$$

The link function $\phi$ is defined as $\phi(z) := f'(z)$ .

Defining the following Lagrange function:

$$\mathcal{L}(\pi, \lambda, \alpha) = -\mathbb{E}_\pi[r^*(c, x_{0:T})] + \beta D_f(\pi_\theta(x_{0:T} \mid c) \parallel \pi_{\text{ref}}(x_{0:T} \mid c)) \tag{27}$$

$$+ \lambda\left(\sum_{x_{0:T}} \pi_\theta(x_{0:T} \mid c) - 1\right) - \sum_{x_{0:T}} \alpha(x_{0:T})\pi_\theta(x_{0:T} \mid c) \tag{28}$$

Employing the Karush-Kuhn-Tucker (KKT) conditions for analysis: Firstly, the stationarity condition necessitates that the gradient of the Lagrangian function with respect to the primal variables should be zero:

Firstly, the stationarity condition requires that the gradient of the Lagrangian with respect to each original variable be zero at the optimal solution.

$$\nabla_{\pi_\theta(x_{0:T}|c)} L(\pi, \lambda, \alpha) = 0 \quad \forall x_{0:T}. \tag{29}$$

By setting the derivative of the Lagrangian with respect to $\pi(y \mid x)$ to zero and further derivation, we can get:

$$r(c, x_{0:T}) - \beta\phi(\frac{\pi_\theta^\star(x_{0:T} \mid c)}{\pi_{\text{ref}}(x_{0:T} \mid c)}) - \lambda + \alpha(x_{0:T}) = 0 \tag{30}$$

The primal feasibility condition requires that the solution satisfies all the original constraints.

$$\text{s.t.} \sum_{x_{0:T}} \pi_\theta(x_{0:T} \mid c) = 1, \pi_\theta(x_{0:T} \mid c) \geq 0 \quad \forall x_{0:T}. \tag{31}$$

Dual feasibility requires that the Lagrange multipliers corresponding to the inequality constraints must be non-negative to ensure that the dual problem remains valid and feasible, preventing negative importance from being assigned to any inequality constraint and maintaining the consistency and correctness of both the primal and dual formulations.

$$\alpha(x_{0:T}) \geq 0 \quad \forall x_{0:T} \tag{32}$$

Complementary slackness requires that for each inequality constraint, either the constraint is satisfied exactly as an equality or its corresponding Lagrange multiplier is zero, ensuring that only active constraints influence the objective function, while inactive constraints are effectively excluded from affecting the solution.

$$\alpha(x_{0:T})\pi_\theta(x_{0:T} \mid c) = 0 \quad \forall x_{0:T} \tag{33}$$

Since $0 \notin \mathrm{dom}(f')$, this ensures that $\frac{\pi(a|s)}{\pi_{\mathrm{ref}}(a|s)}$ is always positive. Assuming that the reference policy satisfies the condition $\pi_{\mathrm{ref}}(a \mid s) > 0$, it follows that $\pi(a \mid s)$ must also be greater than 0. Therefore, based on the above analysis, we arrive at the following conclusion:

$$\alpha(x_{0:T}) = 0 \quad \forall x_{0:T} \tag{34}$$

Incorporating the above conclusion into the stationarity condition results in:

$$r(c, x_{0:T}) - \beta\phi(\frac{\pi_\theta^\star(x_{0:T} \mid c)}{\pi_{\mathrm{ref}}(x_{0:T} \mid c)}) - \lambda = 0 \tag{35}$$

By performing some algebraic manipulations, we obtain:

$$r(c, x_{0:T}) = \beta\phi(\frac{\pi_\theta^\star(x_{0:T} \mid c)}{\pi_{\mathrm{ref}}(x_{0:T} \mid c)}) + \lambda \tag{36}$$

Substituting this expression into the definition of $r(c, x_0) = \mathbb{E}_{\pi_\theta(x_{1:T}|x_0,c)}[r(c, x_{0:T})]$, we obtain the following result.

$$r(c, x_0) = \beta\mathbb{E}_{\pi_\theta(x_{1:T}|x_0,c)}\left[\phi(\frac{\pi_\theta^\star(x_{0:T} \mid c)}{\pi_{\mathrm{ref}}(x_{0:T} \mid c)})\right] + const \tag{37}$$

We observe that the constant const in the formula is unaffected by $x_{0:T}$, ensuring it is canceled out in the Bradley-Terry model. Hence, the proof is complete.

## C. Details of the Primary Derivation

**Lemma 2** Starting from Equation Eq. (11), we derive the following:

$$\mathcal{L}(\theta) = -\log\sigma\left(\beta\mathbb{E}_{\substack{x_{1:T}^+ \sim \pi_\theta(x_{1:T}^+|x_0^+),\\ x_{1:T}^- \sim \pi_\theta(x_{1:T}^-|x_0^-)}}\left[\phi(\frac{\pi_\theta^\star(x_{0:T}^+ \mid c)}{\pi_{\mathrm{ref}}(x_{0:T}^+ \mid c)}) - \phi(\frac{\pi_\theta^\star(x_{0:T}^- \mid c)}{\pi_{\mathrm{ref}}(x_{0:T}^- \mid c)})\right]\right)$$
$$\leq -\mathbb{E}_t\mathbb{E}_{\substack{x_{t-1,t}^+ \sim q(x_{t-1,t}|x_0^+),\\ x_{t-1,t}^- \sim q(x_{t-1,t}|x_0^+)}}\log\sigma\left(\beta T\left[\phi\left(\frac{\pi_\theta^\star(x_{t-1}^+ \mid x_t^+, c)}{\pi_{\mathrm{ref}}(x_{t-1}^+ \mid x_t^+, c)}\right) - \phi\left(\frac{\pi_\theta^\star(x_{t-1}^- \mid x_t^-, c)}{\pi_{\mathrm{ref}}(x_{t-1}^- \mid x_t^-, c)}\right)\right]\right) \tag{38}$$

Proof. By substituting this reward reparameterization into the maximum likelihood objective of the Bradley-Terry model as shown in Eq. (4), the partition function cancels for image pairs, resulting in a maximum likelihood objective defined on diffusion models. Its per-example formula is:

$$\mathcal{L}(\theta) = -\log\sigma\left(\beta\mathbb{E}_{\substack{x_{1:T}^+ \sim \pi_\theta(x_{1:T}^+|x_0^+),\\ x_{1:T}^- \sim \pi_\theta(x_{1:T}^-|x_0^-)}}\left[\phi(\frac{\pi_\theta^\star(x_{0:T}^+ \mid c)}{\pi_{\mathrm{ref}}(x_{0:T}^+ \mid c)}) - \phi(\frac{\pi_\theta^\star(x_{0:T}^- \mid c)}{\pi_{\mathrm{ref}}(x_{0:T}^- \mid c)})\right]\right)$$
$$= -\log\sigma\left(\beta\mathbb{E}_{\substack{x_{1:T}^+ \sim \pi_\theta(x_{1:T}^+|x_0^+),\\ x_{1:T}^- \sim \pi_\theta(x_{1:T}^-|x_0^-)}}\left[\frac{\pi_\theta(x_{0:T} \mid c)}{\pi_{\mathrm{ref}}(x_{0:T} \mid c)} + \log\left(\frac{\pi_\theta(x_{0:T} \mid c)}{\pi_{\mathrm{ref}}(x_{0:T} \mid c)}\right) - \frac{\pi_\theta(x_{0:T} \mid c)}{\pi_{\mathrm{ref}}(x_{0:T} \mid c)} + \log\left(\frac{\pi_\theta(x_{0:T} \mid c)}{\pi_{\mathrm{ref}}(x_{0:T} \mid c)}\right)\right]\right) \tag{39}$$

Where $x_0^+$ and $x_0^-$ are drawn from a static dataset.

Since sampling from $\pi_\theta(x_{1:T} \mid x_0)$ is computationally infeasible, we adopt $q(x_{1:T} \mid x_0)$ as an approximation.

$$
\begin{aligned}
\mathcal{L}_1(\theta) &= -\log \sigma \left( \beta\, \mathbb{E}_{\substack{x_{1:T}^+ \sim q(x_{1:T}^+|x_0^+) \\ x_{1:T}^- \sim q(x_{1:T}^-|x_0^-)}} \left[ \frac{\pi_\theta(x_{0:T}^+ \mid c)}{\pi_{\mathrm{ref}}(x_{0:T}^+ \mid c)} + \log \frac{\pi_\theta(x_{0:T}^+ \mid c)}{\pi_{\mathrm{ref}}(x_{0:T}^+ \mid c)} - \frac{\pi_\theta^\star(x_{0:T}^- \mid c)}{\pi_{\mathrm{ref}}(x_{0:T}^- \mid c)} - \log \frac{\pi_\theta(x_{0:T}^- \mid c)}{\pi_{\mathrm{ref}}(x_{0:T}^- \mid c)} \right] \right) \\[2mm]
&= -\log \sigma \left( \beta\, \mathbb{E}_{\substack{x_{1:T}^+ \sim q(x_{1:T}^+|x_0^+) \\ x_{1:T}^- \sim q(x_{1:T}^-|x_0^-)}} \left[ \frac{\pi_\theta(x_{0:T}^+ \mid c)}{\pi_{\mathrm{ref}}(x_{0:T}^+ \mid c)} + \log \frac{\pi_\theta(x_{0:T}^+ \mid c)}{\pi_{\mathrm{ref}}(x_{0:T}^+ \mid c)} - \frac{\pi_\theta^\star(x_{0:T}^- \mid c)}{\pi_{\mathrm{ref}}(x_{0:T}^- \mid c)} - \log \frac{\pi_\theta(x_{0:T}^- \mid c)}{\pi_{\mathrm{ref}}(x_{0:T}^- \mid c)} \right] \right) \\[2mm]
&= -\log \sigma \left( \beta\, \mathbb{E}_{\substack{x_{1:T}^+ \sim q(x_{1:T}^+|x_0^+) \\ x_{1:T}^- \sim q(x_{1:T}^-|x_0^-)}} \left[ \exp\left( \log \prod_i \frac{\pi_\theta^\star(x_{t-1}^+ \mid x_t^+, c)}{\pi_{\mathrm{ref}}(x_{t-1}^+ \mid x_t^+, c)} \right) + \log \prod_i \frac{\pi_\theta(x_{t-1}^+ \mid x_t^+, c)}{\pi_{\mathrm{ref}}(x_{t-1}^+ \mid x_t^+, c)} \right.\right. \\[2mm]
&\quad \left.\left. - \exp\left( \log \prod_i \frac{\pi_\theta(x_{t-1}^- \mid x_t^-, c)}{\pi_{\mathrm{ref}}(x^+{}_{t-1}^- \mid x_t^-, c)} \right) - \log \prod_i \frac{\pi_\theta(x_{t-1}^- \mid x_t^-, c)}{\pi_{\mathrm{ref}}(x_{t-1}^- \mid x_t^-, c)} \right] \right) \\[2mm]
&= -\log \sigma \left( \beta\, \mathbb{E}_{\substack{x_{1:T}^+ \sim q(x_{1:T}^+|x_0^+) \\ x_{1:T}^- \sim q(x_{1:T}^-|x_0^-)}} \left[ \exp\left( \sum_{t=1}^T \log \frac{\pi_\theta^\star(x_{t-1}^+ \mid x_t^+, c)}{\pi_{\mathrm{ref}}(x_{t-1}^+ \mid x_t^+, c)} \right) + \sum_{t=1}^T \log \frac{\pi_\theta(x_{t-1}^+ \mid x_t^+, c)}{\pi_{\mathrm{ref}}(x_{t-1}^+ \mid x_t^+, c)} \right.\right. \\[2mm]
&\quad \left.\left. - \exp\left( \sum_{t=1}^T \log \frac{\pi_\theta(x_{t-1}^- \mid x_t^-, c)}{\pi_{\mathrm{ref}}(x^+{}_{t-1}^- \mid x_t^-, c)} \right) - \sum_{t=1}^T \log \frac{\pi_\theta(x_{t-1}^- \mid x_t^-, c)}{\pi_{\mathrm{ref}}(x_{t-1}^- \mid x_t^-, c)} \right] \right) \\[2mm]
&= -\log \sigma \left( \beta\, \mathbb{E}_{\substack{x_{1:T}^+ \sim q(x_{1:T}^+|x_0^+) \\ x_{1:T}^- \sim q(x_{1:T}^-|x_0^-)}} \left[ \exp\left( T\mathbb{E}_t \log \frac{\pi_\theta^\star(x_{t-1}^+ \mid x_t^+, c)}{\pi_{\mathrm{ref}}(x_{t-1}^+ \mid x_t^+, c)} \right) + T\mathbb{E}_t \log \frac{\pi_\theta^\star(x_{t-1}^+ \mid x_t^+, c)}{\pi_{\mathrm{ref}}(x_{t-1}^+ \mid x_t^+, c)} \right.\right. \\[2mm]
&\quad \left.\left. - \exp\left( T\mathbb{E}_t \log \frac{\pi_\theta^\star(x_{t-1}^- \mid x_t^-, c)}{\pi_{\mathrm{ref}}(x_{t-1}^- \mid x_t^-, c)} \right) - T\mathbb{E}_t \log \frac{\pi_\theta^\star(x_{t-1}^- \mid x_t^-, c)}{\pi_{\mathrm{ref}}(x_{t-1}^- \mid x_t^-, c)} \right] \right)
\end{aligned}
\tag{40}
$$

Inserting Eq.(52) into Eq.(40) results in:

$$
\begin{aligned}
\mathcal{L}_1(\theta) &\approx -\log \sigma \left( \beta\, \mathbb{E}_{\substack{x_{1:T}^+ \sim q(x_{1:T}^+|x_0^+) \\ x_{1:T}^- \sim q(x_{1:T}^-|x_0^-)}} T\mathbb{E}_t \left[ \log \frac{\pi_\theta^\star(x_{t-1}^+ \mid x_t^+, c)}{\pi_{\mathrm{ref}}(x_{t-1}^+ \mid x_t^+, c)} + \frac{\pi_\theta^\star(x_{t-1}^+ \mid x_t^+, c)}{\pi_{\mathrm{ref}}(x_{t-1}^+ \mid x_t^+, c)} \right.\right. \\[2mm]
&\quad \left.\left. - \log \frac{\pi_\theta^\star(x_{t-1}^- \mid x_t^-, c)}{\pi_{\mathrm{ref}}(x_{t-1}^- \mid x_t^-, c)} - \frac{\pi_\theta^\star(x_{t-1}^- \mid x_t^-, c)}{\pi_{\mathrm{ref}}(x_{t-1}^- \mid x_t^-, c)} \right] \right) \\[2mm]
&= -\log \sigma \left( \beta T\, \mathbb{E}_t \mathbb{E}_{\substack{x_{1:T}^+ \sim q(x_{1:T}^+|x_0^+) \\ x_{1:T}^- \sim q(x_{1:T}^-|x_0^-)}} \left[ \log \frac{\pi_\theta^\star(x_{t-1}^+ \mid x_t^+, c)}{\pi_{\mathrm{ref}}(x_{t-1}^+ \mid x_t^+, c)} + \frac{\pi_\theta^\star(x_{t-1}^+ \mid x_t^+, c)}{\pi_{\mathrm{ref}}(x_{t-1}^+ \mid x_t^+, c)} \right.\right. \\[2mm]
&\quad \left.\left. - \log \frac{\pi_\theta^\star(x_{t-1}^- \mid x_t^-, c)}{\pi_{\mathrm{ref}}(x_{t-1}^- \mid x_t^-, c)} - \frac{\pi_\theta^\star(x_{t-1}^- \mid x_t^-, c)}{\pi_{\mathrm{ref}}(x_{t-1}^- \mid x_t^-, c)} \right] \right) \\[2mm]
&= -\log \sigma \left( \beta T\, \mathbb{E}_t \mathbb{E}_{\substack{x_{t-1,t}^+ \sim q(x_{t-1,t}|x_0^+) \\ x_{t-1,t}^- \sim q(x_{t-1,t}|x_0^+)}} \left[ \log \frac{\pi_\theta^\star(x_{t-1}^+ \mid x_t^+, c)}{\pi_{\mathrm{ref}}(x_{t-1}^+ \mid x_t^+, c)} + \frac{\pi_\theta^\star(x_{t-1}^+ \mid x_t^+, c)}{\pi_{\mathrm{ref}}(x_{t-1}^+ \mid x_t^+, c)} \right.\right. \\[2mm]
&\quad \left.\left. - \log \frac{\pi_\theta^\star(x_{t-1}^- \mid x_t^-, c)}{\pi_{\mathrm{ref}}(x_{t-1}^- \mid x_t^-, c)} - \frac{\pi_\theta^\star(x_{t-1}^- \mid x_t^-, c)}{\pi_{\mathrm{ref}}(x_{t-1}^- \mid x_t^-, c)} \right] \right)
\end{aligned}
\tag{41}
$$

By Jensen's inequality, we have

$$
\begin{aligned}
L_1(\theta) \leq -\mathbb{E}_t \mathbb{E}_{\substack{x^+_{t-1,t} \sim q(x_{t-1,t}|x_0^+) \\ x^-_{t-1,t} \sim q(x_{t-1,t}|x_0^+)}} \log \sigma \Bigg( \beta T \Bigg[ \log \frac{\pi_\theta^\star(x^+_{t-1} \mid x^+_t, c)}{\pi_{\text{ref}}(x^+_{t-1} \mid x^+_t, c)} + \frac{\pi_\theta^\star(x^+_{t-1} \mid x^+_t, c)}{\pi_{\text{ref}}(x^+_{t-1} \mid x^+_t, c)} \\
- \log \frac{\pi_\theta^\star(x^-_{t-1} \mid x^-_t, c)}{\pi_{\text{ref}}(x^-_{t-1} \mid x^-_t, c)} - \frac{\pi_\theta^\star(x^-_{t-1} \mid x^-_t, c)}{\pi_{\text{ref}}(x^-_{t-1} \mid x^-_t, c)} \Bigg] \Bigg) \\
= -\mathbb{E}_t \mathbb{E}_{\substack{x^+_{t-1,t} \sim q(x_{t-1,t}|x_0^+) \\ x^-_{t-1,t} \sim q(x_{t-1,t}|x_0^+)}} \log \sigma \Bigg( \beta T \Bigg[ \phi \left( \frac{\pi_\theta^\star(x^+_{t-1} \mid x^+_t, c)}{\pi_{\text{ref}}(x^+_{t-1} \mid x^+_t, c)} \right) - \phi \left( \frac{\pi_\theta^\star(x^-_{t-1} \mid x^-_t, c)}{\pi_{\text{ref}}(x^-_{t-1} \mid x^-_t, c)} \right) \Bigg] \Bigg)
\end{aligned}
\tag{42}
$$

**Lemma 3** We define the problem under the assumption that two diffusion models $\pi_\theta$ and $\pi_{\text{ref}}$ are available, along with a prompt distribution $p(c)$, a reward function $r(x_0, c)$, and a constant $\beta > 0$. Starting from Equation (6), we derive the following:

$$
- \mathbb{E}_{\mathbf{c} \sim \mathcal{D}_\mathbf{c}, x_0 \sim \pi_\theta(x_0|\mathbf{c})}[r(x, c)] + \beta D_{f_\chi}(\pi \parallel \pi_{\text{ref}}) \tag{43}
$$

$$
= - \mathbb{E}_{\mathbf{c} \sim \mathcal{D}_\mathbf{c}, x_0 \sim \pi_\theta(x_0|\mathbf{c})}[r(c, x_0)] + \beta (D_{\chi^2}(\pi_\theta(x_0 \mid c) \parallel \pi_{\text{ref}}(x_0 \mid c)) + D_{\text{KL}}(\pi_\theta(x_0 \mid c) \parallel \pi_{\text{ref}}(x_0 \mid c))) \tag{44}
$$

$$
\leq - \mathbb{E}_{\mathbf{c} \sim \mathcal{D}_\mathbf{c}, x_0 \sim \pi_\theta(x_{0:T}|\mathbf{c})}[r(c, x_{0:T})] + \beta (D_{\chi^2}(\pi_\theta(x_{0:T} \mid c) \parallel \pi_{\text{ref}}(x_{0:T} \mid c)) + D_{\text{KL}}(\pi_\theta(x_{0:T} \mid c) \parallel \pi_{\text{ref}}(x_{0:T} \mid c))) \tag{45}
$$

$$
\pi(x_0|c) = \int \pi(x_{0:T}|c)\, dx_{1:T} = \int p(x_T) \prod_{t=1}^T \pi(x_{t-1}|x_t, c)\, dx_{1:T}. \tag{46}
$$

Proof. It suffices to show that for any c,

$$
\begin{aligned}
D_f(\pi \parallel \pi_{\text{ref}}) &= D_{\chi^2}(\pi_\theta(x_{0:T} \mid c) \parallel \pi_{\text{ref}}(x_{0:T} \mid c)) + D_{\text{KL}}(\pi_\theta(x_{0:T} \mid c) \parallel \pi_{\text{ref}}(x_{0:T} \mid c)) \tag{47} \\
&\geq D_{\chi^2}(\pi_\theta(x_0 \mid c) \parallel \pi_{\text{ref}}(x_0 \mid c)) + D_{\text{KL}}(\pi_\theta(x_0 \mid c) \parallel \pi_{\text{ref}}(x_0 \mid c)) \tag{48}
\end{aligned}
$$

This can be proved similarly as the data processing inequality. We provide the proof below.

$$
\begin{aligned}
D_f(\pi \parallel \pi_{\text{ref}}) &= D_{\chi^2}(\pi_\theta(x_{0:T} \mid c) \parallel \pi_{\text{ref}}(x_{0:T} \mid c)) + D_{\text{KL}}(\pi_\theta(x_{0:T} \mid c) \parallel \pi_{\text{ref}}(x_{0:T} \mid c)) \\
&= \mathbb{E}_{\pi_\theta(x_{0:T}|c)} \left[ \frac{\pi_\theta(x_{0:T} \mid c)}{\pi_{\text{ref}}(x_{0:T} \mid c)} \right] + \mathbb{E}_{\pi_\theta(x_{0:T}|c)} \left[ \log \frac{\pi_\theta(x_{0:T} \mid c)}{\pi_{\text{ref}}(x_{0:T} \mid c)} \right] \\
&= \int \pi_\theta(x_{0:T} \mid c) \frac{\pi_\theta(x_{0:T} \mid c)}{\pi_{\text{ref}}(x_{0:T} \mid c)} dx_{0:T} + \mathbb{E}_{\pi_\theta(x_0|c)} \left[ \log \frac{\pi_\theta(x_0 \mid c)}{\pi_{\text{ref}}(x_0 \mid x_0, c)} + \log \frac{\pi_\theta(x_{1:T} \mid x_0, c)}{\pi_{\text{ref}}(x_{1:T} \mid x_0, c)} \right] \\
&= \int \frac{\pi_\theta^2(x_{0:T} \mid c)}{\pi_{\text{ref}}(x_{0:T} \mid c)} dx_{0:T} + \mathbb{E}_{\pi_\theta(x_0|c)} \left[ \log \frac{\pi_\theta(x_0 \mid c)}{\pi_{\text{ref}}(x_0 \mid c)} \right] + \mathbb{E}_{\pi_\theta(x_0|c)} \left[ \mathbb{E}_{\pi_\theta(x_{1:T}|x_0,c)} \left[ \log \frac{\pi_\theta(x_{1:T} \mid x_0, c)}{\pi_{\text{ref}}(x_{1:T} \mid x_0, c)} \right] \right] \\
&= \int \frac{\pi_\theta^2(x_{0:T} \mid c)}{\pi_{\text{ref}}(x_{0:T} \mid c)} dx_{0:T} + D_{\text{KL}}(\pi_\theta(x_0 \mid c) \parallel \pi_{\text{ref}}(x_0 \mid c)) \\
&\quad + \mathbb{E}_{\pi_\theta(x_0|c)} [D_{\text{KL}}(\pi_\theta(x_{1:T} \mid x_0, c) \parallel \pi_{\text{ref}}(x_{1:T} \mid x_0, c))]
\end{aligned}
\tag{49}
$$

For non-negative functions $\frac{\pi_\theta^2(x|c)}{\pi_{\text{ref}}(x|c)}$ and a subset $x_{0:T}$ of the domain $x_0$ , we have:

$$
\int \frac{\pi_\theta^2(x_{0:T} \mid c)}{\pi_{\text{ref}}(x_{0:T} \mid c)} dx_{0:T} \geq \int \frac{\pi_\theta^2(x_0 \mid c)}{\pi_{\text{ref}}(x_0 \mid c)} dx_0. \tag{50}
$$

Therefore, the inequality holds:

$$D_{f_{\chi^2}}(\pi \parallel \pi_{\text{ref}}) \geq \int \frac{\pi_\theta^2(x_0 \mid c)}{\pi_{\text{ref}}(x_0 \mid c)} dx_0 + D_{\text{KL}}(\pi_\theta(x_0 \mid c) \parallel \pi_{\text{ref}}(x_0 \mid c)) + \mathbb{E}_{\pi_\theta(x_0\mid c)}\left[D_{\text{KL}}(\pi_\theta(x_{1:T} \mid x_0, c) \parallel \pi_{\text{ref}}(x_{1:T} \mid x_0, c))\right]$$
$$\geq D_{\chi^2}(\pi_\theta(x_0 \mid c) \parallel \pi_{\text{ref}}(x_0 \mid c)) + D_{\text{KL}}(\pi_\theta(x_0 \mid c) \parallel \pi_{\text{ref}}(x_0 \mid c))$$
$$(51)$$

This concludes our proof.

**Lemma 4** Define:
$$\exp\left(T \mathbb{E}_t \log (R_t)\right) \approx T \mathbb{E}_t (R_t) \tag{52}$$

where $R_t = \frac{\pi_\theta^\star(x^+t-1\mid x^+t,c)}{\pi\text{ref}(x^+t-1\mid x_t^+,c)} = 1 + \delta_t$ , with $\delta_t \in [0, 0.1]$

Proof: Expanding $\log(R_t)$ around $R_t = 1$ using a Taylor series, we have

$$\log(R_t) = \log(1 + \delta_t) \approx \delta_t - \frac{\delta_t^2}{2} + \frac{\delta_t^3}{3} - \cdots$$

Given that $\delta_t$ is small, higher-order terms beyond the linear term can be neglected, yielding the approximation

$$\log(R_t) \approx \delta_t$$

The Taylor series expansion of $\exp(x)$ for any real number x is given by

$$\exp(x) = 1 + x + \frac{x^2}{2!} + \frac{x^3}{3!} + \cdots$$

Let $\mathbb{E}_t[\delta_t] = \mu$ , where $\mu \in [0, 0.1]$ , and substitute

$$x = T \mathbb{E}_t[\delta_t] = T \mu$$

Given the small magnitude of $\mu$ , higher-order terms can be neglected, yielding the approximation. $\mathbb{E}_t[\delta_t] = \mu$

$$\exp\left(T \mathbb{E}_t[\log(R_t)]\right) \approx \exp\left(T \mathbb{E}_t[\delta_t]\right) \approx 1 + T \mathbb{E}_t[\delta_t] + \frac{1}{2}\left(T \mathbb{E}_t[\delta_t]\right)^2 \approx 1 + T\mu + \frac{1}{2}(T\mu)^2$$

According to $R_t = 1 + \delta_t$, the expected value is:

$$T \mathbb{E}_t[R_t] = T \mathbb{E}_t[1 + \delta_t] = T (1 + \mathbb{E}_t[\delta_t]) = T + T \mathbb{E}_t[\delta_t] = T + T \mu$$

A rigorous demonstration of the equivalence $\exp\left(T \mathbb{E}_t[\log(R_t)]\right) \approx T \mathbb{E}_t[R_t]$ fundamentally reduces to showing that:

$$1 + T\mu + \frac{1}{2}(T\mu)^2 \approx T + T\mu.$$

where T = 1000,The solution yields $\mu \approx 0.0447$, which lies within the interval [0, 0.1],the original equatio holds approximately true within the defined range of $\mu$ . we obtain the following result

$$\exp\left(T \mathbb{E}_t \log (R_t)\right) \approx T \mathbb{E}_t (R_t) \tag{53}$$

# D. Details of the Primary Derivation

This section details the derivation of the Diffusion-$\chi$PO loss as shown in Eq.(13) , starting from the loss function in Eq.(12):

$$
\begin{aligned}
\mathcal{L}(\theta) \leq &- \mathbb{E}_{\substack{(x_0^+, x_0^-) \sim D, t \sim U(0,T), \\ x_{t-1,t}^+ \sim p_\theta(x_{t-1,t}^+ | x_0^+) \\ x_{t-1,t}^- \sim p_\theta(x_{l,t-1,t}^- | x_0^-)}} \log \sigma \left( T\beta \left[ \phi \left( \frac{\pi_\theta^\star(x_{t-1}^+ \mid x_t^+, c)}{\pi_{\text{ref}}(x_{t-1}^+ \mid x_t^+, c)} \right) - \phi \left( \frac{\pi_\theta^\star(x_{t-1}^- \mid x_t^-, c)}{\pi_{\text{ref}}(x_{t-1}^- \mid x_t^-, c)} \right) \right] \right) \\
= &- \mathbb{E} \log \sigma \left( \beta T \left[ \log \frac{\pi_\theta^\star(x_{t-1}^+ \mid x_t^+, c)}{\pi_{\text{ref}}(x_{t-1}^+ \mid x_t^+, c)} + \frac{\pi_\theta^\star(x_{t-1}^+ \mid x_t^+, c)}{\pi_{\text{ref}}(x_{t-1}^+ \mid x_t^+, c)} - \log \frac{\pi_\theta^\star(x_{t-1}^- \mid x_t^-, c)}{\pi_{\text{ref}}(x_{t-1}^- \mid x_t^-, c)} - \frac{\pi_\theta^\star(x_{t-1}^- \mid x_t^-, c)}{\pi_{\text{ref}}(x_{t-1}^- \mid x_t^-, c)} \right] \right)
\end{aligned}
\tag{54}
$$

Following the approach of (Ho et al., 2020), the policies are defined as:

$$
\begin{aligned}
\pi_\theta(x_{t-1}^* \mid x_t^*) &= \mathcal{N} \left( x_{t-1}^*; \frac{\sqrt{\alpha_{t-1}}}{\sqrt{\alpha_t}} (x_t^* - \frac{\beta_t}{\sqrt{1-\bar{\alpha}_t}} \epsilon_\theta(x_t^*, t)), \sigma_t^2 \right) \\
&= \frac{1}{\left( \sqrt{2\pi\sigma_t^2} \right)^d} \exp \left( -\frac{1}{2\sigma_t^2} \left\| x_{t-1}^* - \sqrt{\frac{\alpha_{t-1}}{\alpha_t}} (x_t^* - \frac{\beta_t}{\sqrt{1-\bar{\alpha}_t}} \epsilon_\theta(x_t^*, t)) \right\|_2^2 \right)
\end{aligned}
$$

In this context, $d$ signifies the dimensionality of the image vector, and we utilize $y_t^*$ to streamline the notation. The subsequent derivation using $y_t^*$ applies to both $y_t^+$ and $y_t^-$ . We can represent the ground-truth denoising distribution and posterior mean in the following form:

$$
\begin{aligned}
q \left( x_{t-1}^* \mid x_t^*, x_0^* \right) &= \mathcal{N}(x_{t-1}^*; \sqrt{\frac{\alpha_{t-1}}{\alpha_t}} (x_t^* - \frac{\beta_t}{\sqrt{1-\bar{\alpha}_t}} \epsilon_t), \sigma_t^2) \\
&= \frac{1}{\left( \sqrt{2\pi\sigma_t^2} \right)^d} \exp \left( -\frac{1}{2\sigma_t^2} \left\| x_{t-1}^* - \sqrt{\frac{\alpha_{t-1}}{\alpha_t}} (x_t^* - \frac{\beta_t}{\sqrt{1-\bar{\alpha}_t}} \epsilon_t) \right\|_2^2 \right) \\
\mathbb{E} \left[ x_{t-1}^* \mid x_t^*, x_0^* \right] &= \sqrt{\frac{\alpha_{t-1}}{\alpha_t}} (x_t^* - \frac{\beta_t}{\sqrt{1-\bar{\alpha}_t}} \epsilon_t)
\end{aligned}
$$

Here, $x_0^*$ is sourced from the offline dataset $D$ , and $x_t^*$ is obtained by sampling from the forward process $q(x_t^* \mid y_0^*)$ .

When $x_{t-1}^*$ is sampled from $q(x_{t-1}^* \mid x_t^*, x_0^*)$ , it can be written as: $x_t^* = \sqrt{\frac{\alpha_{t-1}}{\alpha_t}} \left( x_t^* - \frac{\beta_t}{\sqrt{1-\bar{\alpha}_t}} \epsilon_t \right) + \sigma_t \epsilon_{t-1}$ . In this expression, $\epsilon_t$ is the noise introduced in the forward diffusion process to obtain $x_t^*$ , and $\epsilon_{t-1}$ is the Gaussian noise used to derive $x_{t-1}^*$ in the reverse diffusion process through the re-parametrization trick. The policy evaluation is then conducted as:

$$\pi_\theta(x_{t-1}^*|x_t^*) = \frac{1}{\left(\sqrt{2\pi\sigma_t^2}\right)^d} \exp\left( -\frac{1}{2\sigma_{t-1}^2} \left\| \sqrt{\frac{\alpha_{t-1}}{\alpha_t}}(x_t^* - \frac{\beta_t}{\sqrt{1-\bar\alpha_t}}\boldsymbol\epsilon_t^*) + \sigma_t\boldsymbol\epsilon_{t-1}^* \right.\right.$$

$$\left.\left. -\sqrt{\frac{\alpha_{t-1}}{\alpha_t}}(x_t^* - \frac{\beta_t}{\sqrt{1-\bar\alpha_t}}\boldsymbol\epsilon_\theta(x_t^*,t)) \right\|_2^2 \right)$$

$$= \frac{1}{\left(\sqrt{2\pi\sigma_t^2}\right)^d} \exp\left( -\frac{1}{2\sigma_t^2}\frac{\alpha_{t-1}}{\alpha_t}\frac{\beta_t^2}{1-\bar\alpha_t} \left\| \boldsymbol\epsilon_\theta(x_t^*,t) - \boldsymbol\epsilon_t^* + \sigma_t\boldsymbol\epsilon_{t-1}^* \right\|_2^2 \right)$$

$$= \frac{1}{\left(\sqrt{2\pi\sigma_t^2}\right)^d} \exp\left( -\frac{1}{2}\frac{1-\bar\alpha_t}{(1-\bar\alpha_{t-1})\beta_t}\frac{\alpha_{t-1}}{\alpha_t}\frac{\beta_t^2}{1-\bar\alpha_t} \left\| \boldsymbol\epsilon_\theta(x_t^*,t) - \boldsymbol\epsilon_t^* + \sigma_t\boldsymbol\epsilon_{t-1}^* \right\|_2^2 \right)$$

$$= \frac{1}{\left(\sqrt{2\pi\sigma_t^2}\right)^d} \exp\left( -\frac{1}{2}\frac{\beta_t}{(1-\bar\alpha_{t-1})}\frac{\alpha_{t-1}}{\alpha_t} \left\| \boldsymbol\epsilon_\theta(x_t^*,t) - \boldsymbol\epsilon_t^* + \sigma_t\boldsymbol\epsilon_{t-1}^* \right\|_2^2 \right)$$

The log probability is defined as follows:

$$\log \pi_\theta(x_{t-1}^*|x_t^*) = -\frac{1}{2}\frac{\beta_t\alpha_{t-1}}{(1-\bar\alpha_{t-1})\alpha_t} \left\| \boldsymbol\epsilon_\theta(x_t^*,t) - \boldsymbol\epsilon_t^* + \sigma_t\boldsymbol\epsilon_{t-1}^* \right\|_2^2$$
$$-\frac{d}{2}\cdot\log 2\pi - d\cdot\log\sigma_t \tag{55}$$

Inserting Eq. (55) into Eq. (54) results in:

$$\tilde{\mathcal{L}}(\theta) = -\mathbb{E}\log\sigma\left( -\beta T\left[ \log\frac{\pi_\theta(x_{t-1}^+|x_t^+)}{\pi_{\text{ref}}(x_{t-1}^+|x_t^+)} - \log\frac{\pi_\theta(x_{t-1}^-|x_t^-)}{\pi_{\text{ref}}(x_{t-1}^-|x_t^-)} + \frac{\pi_\theta^\star(x_{t-1}^+\mid x_t^+,c)}{\pi_{\text{ref}}(x_{t-1}^+\mid x_t^+,c)} - \frac{\pi_\theta^\star(x_{t-1}^-\mid x_t^-,c)}{\pi_{\text{ref}}(x_{t-1}^-\mid x_t^-,c)} \right] \right)$$

$$= -\mathbb{E}\log\sigma\left( -\beta T\left[ \log\pi_\theta(x_{t-1}^+|x_t^+) - \log\pi_{\text{ref}}(x_{t-1}^+|x_t^+) - \left( \left(\log\pi_\theta(x_{t-1}^-|x_t^-) - \log\pi_{\text{ref}}(x_{t-1}^-|x_t^-)\right) \right) \right.\right.$$

$$\left.\left. + \frac{\pi_\theta^\star(x_{t-1}^+\mid x_t^+,c)}{\pi_{\text{ref}}(x_{t-1}^+\mid x_t^+,c)} - \frac{\pi_\theta^\star(x_{t-1}^-\mid x_t^-,c)}{\pi_{\text{ref}}(x_{t-1}^-\mid x_t^-,c)} \right] \right)$$

$$= -\mathbb{E}\log\sigma\left( -\beta T\left[ \frac{\beta_t\alpha_{t-1}}{2(1-\bar\alpha_{t-1})\alpha_t}\left( \left\| \boldsymbol\epsilon_\theta(x_t^+,t) - \boldsymbol\epsilon_t^+ + \sigma_t\boldsymbol\epsilon_{t-1}^+ \right\|_2^2 - \left\| \boldsymbol\epsilon_{\text{ref}}(x_t^+,t) - \boldsymbol\epsilon_t^+ + \sigma_t\boldsymbol\epsilon_{t-1}^+ \right\|_2^2 \right.\right.\right.$$

$$\left. -\left( \left\| \boldsymbol\epsilon_\theta(x_t^-,t-1) - \boldsymbol\epsilon_t^- + \sigma_t\boldsymbol\epsilon_{t-1}^- \right\|_2^2 - \left\| \boldsymbol\epsilon_{\text{ref}}(x_t^-,t-1) - \boldsymbol\epsilon_t^- + \sigma_t\boldsymbol\epsilon_{t-1}^- \right\|_2^2 \right) \right) +$$

$$\exp\left( \frac{\beta_t\alpha_{t-1}}{2(1-\bar\alpha_{t-1})\alpha_t}(\left\| \boldsymbol\epsilon_\theta(x_t^+,t) - \boldsymbol\epsilon_t^+ + \sigma_t\boldsymbol\epsilon_{t-1}^+ \right\|_2^2 - \left\| \boldsymbol\epsilon_{\text{ref}}(x_t^+,t) - \boldsymbol\epsilon_t^+ + \sigma_t\boldsymbol\epsilon_{t-1}^+ \right\|_2^2) \right)$$

$$\left.\left. -\exp\left( \frac{\beta_t\alpha_{t-1}}{2(1-\bar\alpha_{t-1})\alpha_t}(\left\| \boldsymbol\epsilon_\theta(x_t^-,t) - \boldsymbol\epsilon_t^- + \sigma_t\boldsymbol\epsilon_{t-1}^- \right\|_2^2 - \left\| \boldsymbol\epsilon_{\text{ref}}(x_t^-,t) - \boldsymbol\epsilon_t^- + \sigma_t\boldsymbol\epsilon_{t-1}^- \right\|_2^2) \right) \right]\right)$$

Similarly, by approximating $x_{t-1}^*$ with the mean of $q(x_{t-1}^*\mid x_t^*,x_0^*)$, the policy is assessed as follows:

$$\pi_\theta(\mathbb{E}\left[x_{t-1}^* \mid x_t^*, x_0^*\right] \mid x_t^*) = \frac{1}{\left(\sqrt{2\pi\sigma_t^2}\right)^d} \exp\left(-\frac{1}{2\sigma_{t-1}^2}\left\|\sqrt{\frac{\alpha_{t-1}}{\alpha_t}}(x_t^* - \frac{\beta_t}{\sqrt{1-\bar{\alpha}_t}}\boldsymbol{\epsilon}_t^*)\right.\right.$$

$$\left.\left. - \sqrt{\frac{\alpha_{t-1}}{\alpha_t}}(x_t^* - \frac{\beta_t}{\sqrt{1-\bar{\alpha}_t}}\boldsymbol{\epsilon}_\theta(x_t^*, t))\right\|_2^2\right)$$

$$= \frac{1}{\left(\sqrt{2\pi\sigma_t^2}\right)^d} \exp\left(-\frac{1}{2\sigma_t^2}\frac{\alpha_{t-1}}{\alpha_t}\frac{\beta_t^2}{1-\bar{\alpha}_t}\left\|\boldsymbol{\epsilon}_\theta(x_t^*, t) - \boldsymbol{\epsilon}_t^*\right\|_2^2\right)$$

$$= \frac{1}{\left(\sqrt{2\pi\sigma_t^2}\right)^d} \exp\left(-\frac{1}{2}\frac{1-\bar{\alpha}_t}{(1-\bar{\alpha}_{t-1})\beta_t}\frac{\alpha_{t-1}}{\alpha_t}\frac{\beta_t^2}{1-\bar{\alpha}_t}\left\|\boldsymbol{\epsilon}_\theta(x_t^*, t) - \boldsymbol{\epsilon}_t^*\right\|_2^2\right)$$

$$= \frac{1}{\left(\sqrt{2\pi\sigma_t^2}\right)^d} \exp\left(-\frac{1}{2}\frac{\beta_t}{(1-\bar{\alpha}_{t-1})}\frac{\alpha_{t-1}}{\alpha_t}\left\|\boldsymbol{\epsilon}_\theta(x_t^*, t) - \boldsymbol{\epsilon}_t^*\right\|_2^2\right)$$

Once more, the log probability is defined as:

$$\log \pi_\theta(\mathbb{E}\left[x_{t-1}^* \mid x_t^*, x_0^*\right] \mid x_t^*) = -\frac{1}{2}\frac{\beta_t\alpha_{t-1}}{(1-\bar{\alpha}_{t-1})\alpha_t}\left\|\boldsymbol{\epsilon}_\theta(x_t^*, t) - \boldsymbol{\epsilon}_t^*\right\|_2^2$$

$$- \frac{d}{2}\cdot\log 2\pi - d\cdot\log\sigma_t \tag{56}$$

Inserting Eq. (56) into Eq. (54) results in:

$$\hat{\mathcal{L}}(\theta) = -\mathbb{E}\log\sigma\left(-\beta T\left[\log\frac{\pi_\theta(\mathbb{E}\left[x_{t-1}^+ \mid x_t^+, x_0^+\right] \mid x_t^+)}{\pi_{\text{ref}}(\mathbb{E}\left[x_{t-1}^+ \mid x_t^+, x_0^+\right] \mid x_t^+)} - \log\frac{\pi_\theta(\mathbb{E}\left[x_{t-1}^- \mid x_t^-, x_0^-\right] \mid x_t^-)}{\pi_{\text{ref}}(\mathbb{E}\left[x_{t-1}^- \mid x_t^-, x_0^-\right] \mid x_t^-)}\right.\right.$$

$$\left.\left. + \frac{\pi_\theta^\star(x_{t-1}^+ \mid x_t^+, c)}{\pi_{\text{ref}}(x_{t-1}^+ \mid x_t^+, c)} - \frac{\pi_\theta^\star(x_{t-1}^- \mid x_t^-, c)}{\pi_{\text{ref}}(x_{t-1}^- \mid x_t^-, c)}\right]\right)$$

$$= -\mathbb{E}\log\sigma\left(-\beta T\left[\log\pi_\theta(\mathbb{E}\left[x_{t-1}^+ \mid x_t^+, x_0^+\right] \mid x_t^+) - \log\pi_{\text{ref}}(\mathbb{E}\left[x_{t-1}^+ \mid x_t^+, x_0^+\right] \mid x_t^+)\right.\right.$$

$$- \log\pi_\theta(\mathbb{E}\left[x_{t-1}^- \mid x_t^-, x_0^-\right] \mid x_t^-) + \log\pi_{\text{ref}}(\mathbb{E}\left[x_{t-1}^- \mid x_t^-, x_0^-\right] \mid x_t^-)$$

$$\left.\left. + \frac{\pi_\theta^\star(x_{t-1}^+ \mid x_t^+, c)}{\pi_{\text{ref}}(x_{t-1}^+ \mid x_t^+, c)} - \frac{\pi_\theta^\star(x_{t-1}^- \mid x_t^-, c)}{\pi_{\text{ref}}(x_{t-1}^- \mid x_t^-, c)}\right]\right)$$

$$= -\mathbb{E}\log\sigma\left(-\beta T\left[\frac{\beta_t\alpha_{t-1}}{2(1-\bar{\alpha}_{t-1})\alpha_t}\left(\left\|\boldsymbol{\epsilon}_\theta(x_t^+, t) - \boldsymbol{\epsilon}_t^+\right\|_2^2 - \left\|\boldsymbol{\epsilon}_{\text{ref}}(x_t^+, t) - \boldsymbol{\epsilon}_t^+\right\|_2^2\right.\right.\right.$$

$$-\left(\left\|\boldsymbol{\epsilon}_\theta(x_t^-, t) - \boldsymbol{\epsilon}_t^-\right\|_2^2 - \left\|\boldsymbol{\epsilon}_{\text{ref}}(x_t^-, t) - \boldsymbol{\epsilon}_t^-\right\|_2^2\right)\right)$$

$$+ \exp\left(\frac{\beta_t\alpha_{t-1}}{2(1-\bar{\alpha}_{t-1})\alpha_t}(\left\|\boldsymbol{\epsilon}_\theta(x_t^+, t) - \boldsymbol{\epsilon}_t^+\right\|_2^2 - \left\|\boldsymbol{\epsilon}_{\text{ref}}(x_t^+, t) - \boldsymbol{\epsilon}_t^+\right\|_2^2)\right)$$

$$\left.\left.- \exp\left(\frac{\beta_t\alpha_{t-1}}{2(1-\bar{\alpha}_{t-1})\alpha_t}(\left\|\boldsymbol{\epsilon}_\theta(x_t^-, t) - \boldsymbol{\epsilon}_t^-\right\|_2^2 - \left\|\boldsymbol{\epsilon}_{\text{ref}}(x_t^-, t) - \boldsymbol{\epsilon}_t^-\right\|_2^2)\right)\right]\right)$$

$$= -\mathbb{E}\log\sigma\left(-\beta T\left[\phi\left(\exp\left(\frac{\beta_t\alpha_{t-1}}{2(1-\bar{\alpha}_{t-1})\alpha_t}(\left\|\boldsymbol{\epsilon}_\theta(x_t^+, t) - \boldsymbol{\epsilon}_t^+\right\|_2^2 - \left\|\boldsymbol{\epsilon}_{\text{ref}}(x_t^+, t) - \boldsymbol{\epsilon}_t^+\right\|_2^2)\right)\right)\right.\right.$$

$$\left.\left.- \phi\left(\exp\left(\frac{\beta_t\alpha_{t-1}}{2(1-\bar{\alpha}_{t-1})\alpha_t}(\left\|\boldsymbol{\epsilon}_\theta(x_t^-, t) - \boldsymbol{\epsilon}_t^-\right\|_2^2 - \left\|\boldsymbol{\epsilon}_{\text{ref}}(x_t^-, t) - \boldsymbol{\epsilon}_t^-\right\|_2^2)\right)\right)\right]\right)$$

# E. Further Analysis on the Gradient Fields

**Lemma 5** The partial derivatives (gradients) of $X_1$ and $X_2$ resulting from Eq.(21) can be expressed as follows:

$$\begin{cases} \frac{\partial L_\phi(Z_1, Z_2)}{\partial Z_1} = -\beta \left(1 - \sigma \left(\beta \phi(Z_1) - \beta \phi(Z_2)\right)\right) \phi'(Z_1) \\ \frac{\partial L_\phi(Z_1, Z_2)}{\partial Z_2} = \beta \left(1 - \sigma \left(\beta \phi(Z_1) - \beta \phi(Z_2)\right)\right) \phi'(Z_2) \end{cases} \tag{57}$$

Consequently, the gradient ratio of $\mathcal{L}_\phi(Z_1, Z_2)$ simplifies to:

$$\left| \frac{\partial \mathcal{L}_\phi(Z_1, Z_2)}{\partial Z_1} \middle/ \frac{\partial \mathcal{L}_\phi(Z_1, Z_2)}{\partial Z_2} \right| = \frac{\phi'(Z_1)}{\phi'(Z_2)} \tag{58}$$

If the regularization link function is $\phi_{\text{DPO}}$, then Eq (57) simplifies to:

$$\begin{cases} \frac{\partial \mathcal{L}_{\phi_{DPO}}(X_1, X_2)}{\partial X_1} = -\beta \frac{X_2^\beta}{X_1 \cdot \left(X_1^\beta + X_2^\beta\right)} \\ \frac{\partial \mathcal{L}_{\phi_{DPO}}(X_1, X_2)}{\partial X_2} = \beta \frac{X_2^{\beta-1}}{\left(X_1^\beta + X_2^\beta\right)} \end{cases} \tag{59}$$

Consequently, the gradient ratio becomes:

$$\left| \frac{\partial \mathcal{L}_{\phi_{DPO}}(Z_1, Z_2)}{\partial Z_1} \middle/ \frac{\partial \mathcal{L}_{\phi_{DPO}}(Z_1, Z_2)}{\partial Z_2} \right| = \frac{Z_2}{Z_1} \tag{60}$$

If the regularization link function is $\phi_\chi$, then Eq (57) simplifies to:

$$\begin{cases} \frac{\partial \mathcal{L}_{\phi_\chi}(Z_1, Z_2)}{\partial Z_1} = -\beta \cdot \frac{Z_1 + 1}{Z_1} \cdot \frac{e^{\beta(Z_2 + \log(Z_2))}}{e^{\beta(Z_1 + \log(Z_1))} + e^{\beta(Z_2 + \log(Z_2))}} \\ \frac{\partial \mathcal{L}_{\phi_\chi}(Z_1, Z_2)}{\partial Z_2} = \beta \cdot (Z_2 + 1) \cdot \frac{e^{\beta Z_2} \cdot Z_2^{\beta-1}}{e^{\beta(Z_1 + \log(Z_1))} + e^{\beta(Z_2 + \log(Z_2))}} \end{cases} \tag{61}$$

Consequently, the gradient ratio becomes:

$$\left| \frac{\partial \mathcal{L}_{\phi_\chi}(Z_1, Z_2)}{\partial Z_1} \middle/ \frac{\partial \mathcal{L}_{\phi_\chi}(Z_1, Z_2)}{\partial Z_2} \right| = \frac{Z_2(Z_1 + 1)}{Z_1(Z_2 + 1} \tag{62}$$

If the regularization link function is $\phi_{\chi^n}$, then Eq (57) simplifies to:

$$\begin{cases} \frac{\partial \mathcal{L}_{\phi_{\chi^n}}(Z_1, Z_2)}{\partial Z_1} = -\beta \cdot \frac{1}{n}(\sum_{k=0}^n Z_1^{k-1}) \cdot \frac{e^{\beta(\frac{1}{n}(\sum_{k=1}^n \frac{1}{k} Z_2^k + \log Z_2))}}{e^{\beta(\frac{1}{n}(\sum_{k=1}^n \frac{1}{k} Z_1^k + \log Z_1))} + e^{\beta(\frac{1}{n}(\sum_{k=1}^n \frac{1}{k} Z_2^k + \log Z_2))}} \\ \frac{\partial \mathcal{L}_{\phi_{\chi^n}}(Z_1, Z_2)}{\partial Z_2} = \beta \cdot \frac{1}{n}(\sum_{k=0}^n Z_2^{k-1}) \cdot \frac{e^{\beta(\frac{1}{n}(\sum_{k=1}^n \frac{1}{k} Z_2^k + \log Z_2))}}{e^{\beta(\frac{1}{n}(\sum_{k=1}^n \frac{1}{k} Z_1^k + \log Z_1))} + e^{\beta(\frac{1}{n}(\sum_{k=1}^n \frac{1}{k} Z_2^k + \log Z_2))}} \end{cases} \tag{63}$$

Consequently, the gradient ratio becomes:

$$\left| \frac{\partial \mathcal{L}_{\phi_{\chi^n}}(Z_1, Z_2)}{\partial Z_1} \middle/ \frac{\partial \mathcal{L}_{\phi_{\chi^n}}(Z_1, Z_2)}{\partial Z_2} \right| = \frac{\sum_{k=0}^n Z_1^{k-1}}{\sum_{k=0}^n Z_2^{k-1}} \tag{64}$$

## F. Pseudocode for Training Objective

```python
def loss(model, ref_model, x_w, x_l, c, beta):
    """
    This is an example psuedo-code snippet for calculating the Diffusion-xnpo losson a single
    model: Diffusion model that accepts prompt conditioning c and time conditioning t
    ref_model: Frozen initialization of model
    x_w: Preferred Image (latents in this work)
    x_l: Non-Preferred Image (latents in this work)
    c: Conditioning (text in this work)
    beta: Regularization Parameter
    xn:  n denotes the exponent of  fxn .
    returns: x^nPO loss value
    """
    timestep = torch.randint(0, 1000)

    noise = torch.randn_like(x_w)

    noisy_x_w = add_noise(x_w, noise, t)

    noisy_x_l = add_noise(x_l, noise, t)

    model_w_pred = model(noisy_x_w, c, t)

    model_l_pred = model(noisy_x_l, c, t)

    ref_w_pred = ref(noisy_x_w, c, t)

    ref_l_pred = ref(noisy_x_l, c, t)

    model_w_err = (model_w_pred - noise).norm().pow(2)

    model_l_err = (model_l_pred - noise).norm().pow(2)

    ref_w_err = (ref_w_pred - noise).norm().pow(2)

    ref_l_err = (ref_l_pred - noise).norm().pow(2)

    weights = [0.5 + 0.5 * i for i in range(xn)]

    weighted_sum = 0.0

    for i, weight in enumerate(weights, start=1):
        exp_w = torch.exp(weight * (model_w_err - ref_w_err)) / i
        exp_l = torch.exp(weight * (model_l_err - ref_l_err)) / i
        weighted_sum += exp_w - exp_l

    weighted_sum += 0.5 * (model_w_err - ref_w_err -model_l_err + ref_l_err)
    inside_term =  - beta * weighted_sum / len(weights)

    loss = -1 * log(sigmoid(inside_term))

    return loss
```

# G. More Images from the Multiple Prompt Experiment

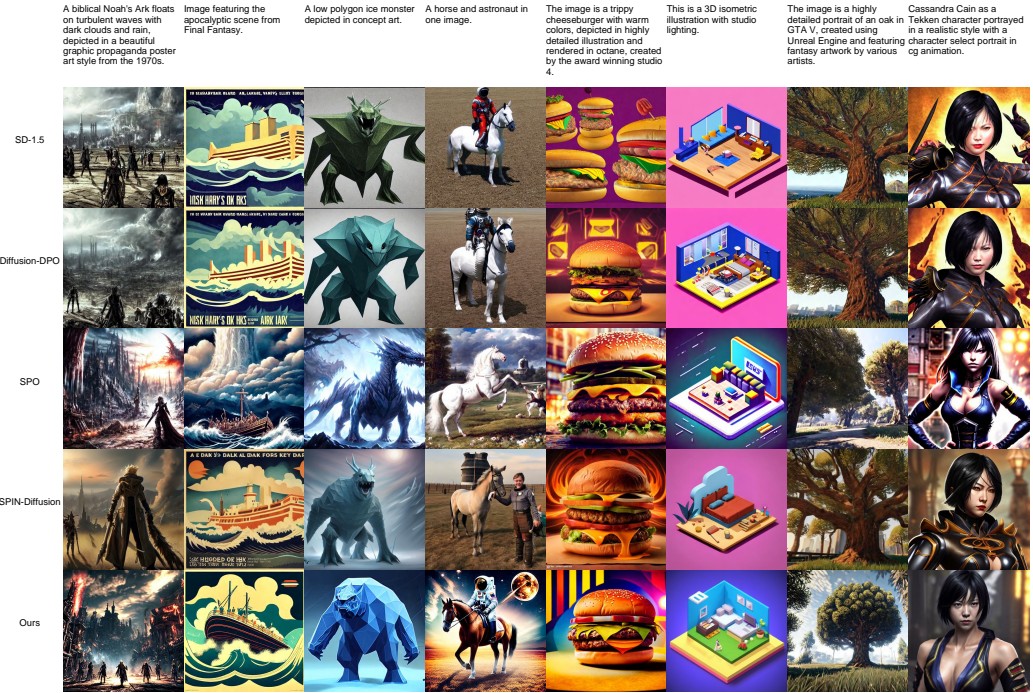

*Figure 4.* Additional Prompt Experiment: This experiment evaluates our method against baseline methods by generating images on the HPSv2 test set prompts under random sampling.

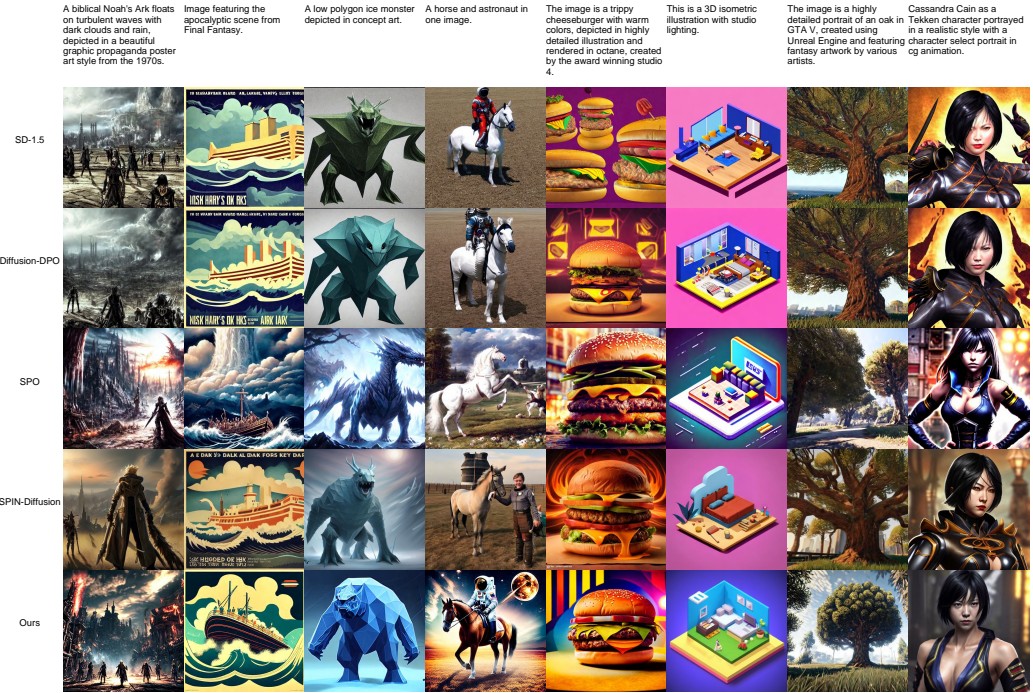

*Figure 5.* Additional Prompt Experiment: This experiment evaluates our method against baseline methods by generating images on the HPSv2 test set prompts under random sampling.

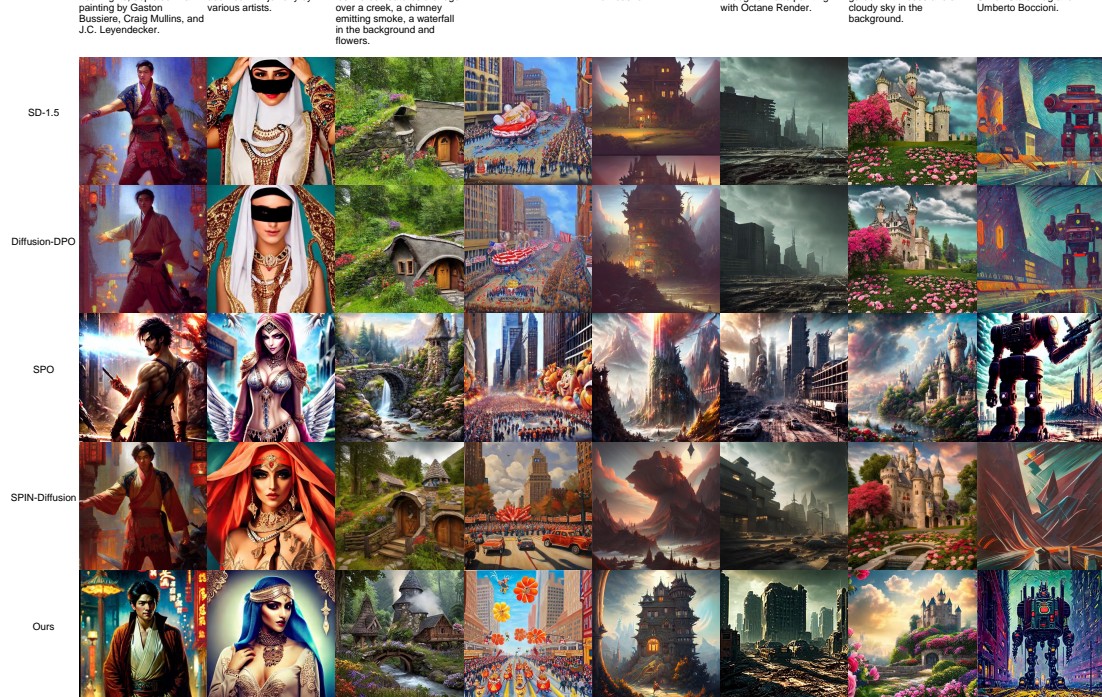

*Figure 6.* Additional Prompt Experiment: This experiment evaluates our method against baseline methods by generating images on the HPSv2 test set prompts under random sampling.

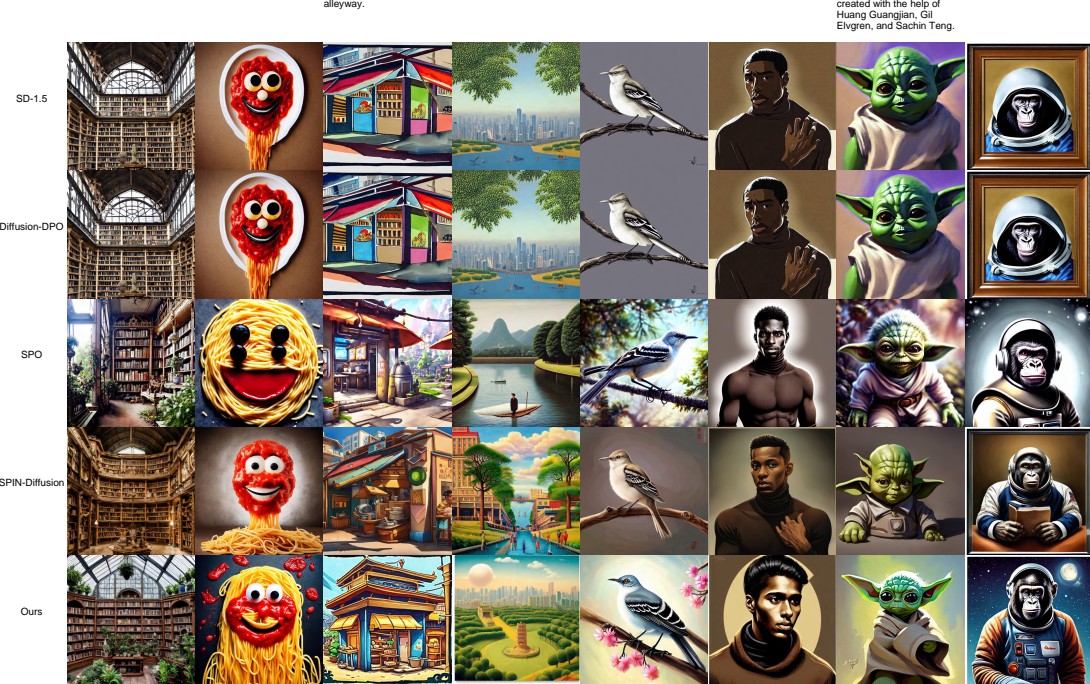

*Figure 7.* Additional Prompt Experiment: This experiment evaluates our method against baseline methods by generating images on the HPSv2 test set prompts under random sampling.

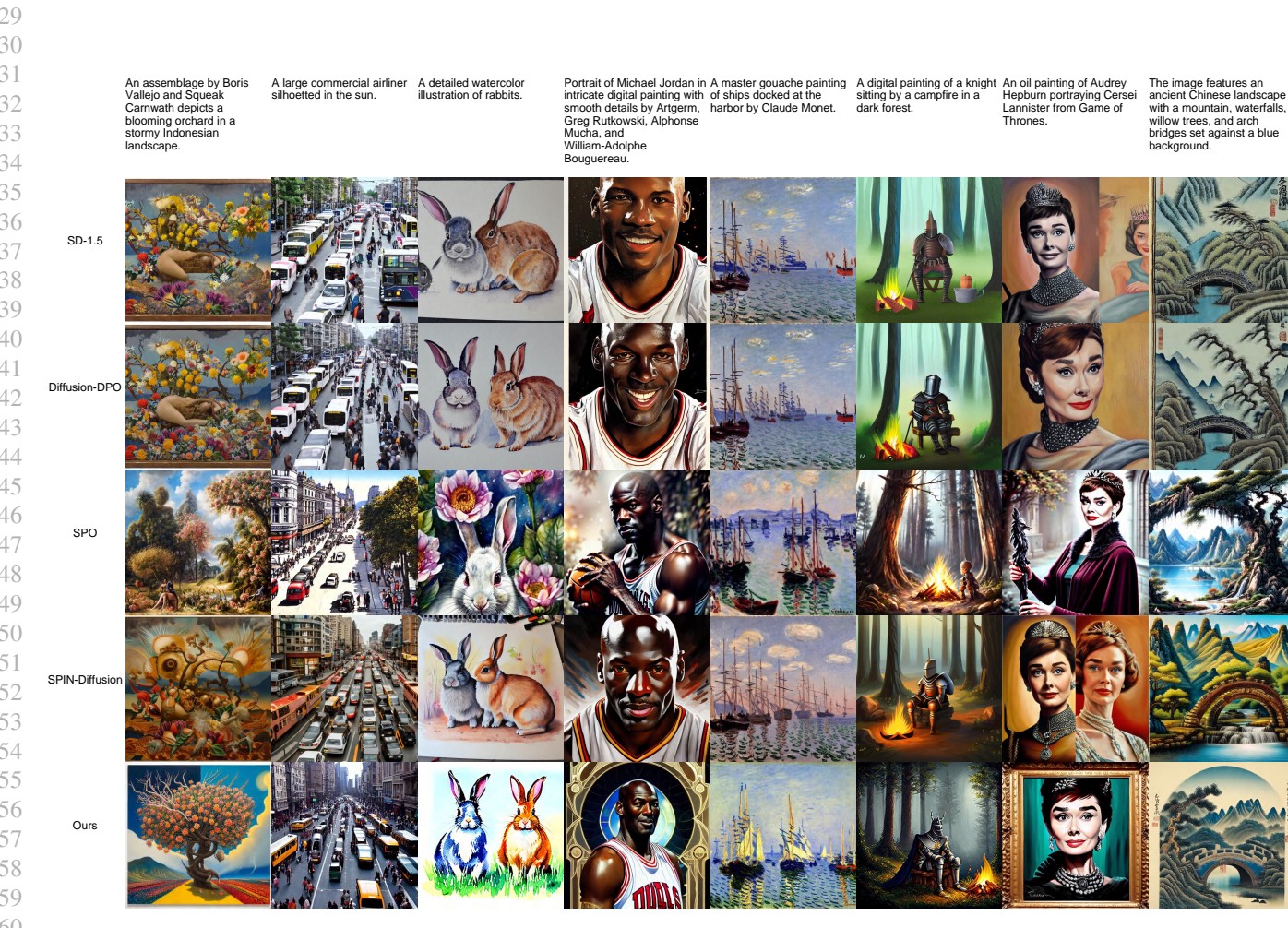

*Figure 8.* Additional Prompt Experiment: This experiment evaluates our method against baseline methods by generating images on the HPSv2 test set prompts under random sampling.

