# OpenReview forum: "Beyond KL-Regularization: Achieving Unbiased Direct Alignment through Diffusion $f_{\chi^n}$-Preference Optimization"
_ICML.cc/2025/Conference — Submitted to ICML 2025_

### Official Review · Reviewer_PJfL · 2025-03-10

**Overall Recommendation:** 2

**Summary:**

The paper presents Diffusion-$\chi^n$PO, a novel method for aligning diffusion models with human preferences in text-to-image generation. It introduces an $f_{\chi^n}$-regularization technique to refine the gradient ratio of the objective function, balancing optimization between preferred and non-preferred samples. The method integrates $f_{\chi^n}$-Preference Optimization ($\chi$PO) into diffusion models, proposing a generalized $f_{\chi^n}$-Preference Optimization ($\chi$PO) framework that enhances flexibility in implicit reward model design and mitigates the impact of conflicting data. Experiments on the Pick-a-Pic dataset demonstrate improved alignment with textual prompts and enhanced visual quality compared to existing methods. The main contributions include the derivation of a stable and efficient loss function for diffusion models, the proposal of the $\chi$PO framework, and an analysis of gradient fields' impacts on alignment.

**Claims And Evidence:**

The claims made in the submission are generally supported by clear and convincing evidence. The paper provides a detailed explanation of the proposed method, including the mathematical derivation of the loss function and the theoretical underpinnings of the $f_{\chi^n}$-regularization technique. The authors also present a thorough analysis of the gradient fields' impacts on the alignment process, which adds credibility to their claims.

**Essential References Not Discussed:**

Although the authors discussed various link functions in Section 4.3, they should provide more analysis on the advantages of $\chi^n$PO  compared to common divergences, such as those discussed in [1]. Additionally, the writing approach may pose reading difficulties for readers unfamiliar with $f_{\chi^n}$-regularization , so more discussion on this topic is recommended.

[1] Wang, C., Jiang, Y., Yang, C., Liu, H., and Chen, Y. Beyond reverse kl: Generalizing direct preference optimization with diverse divergence constraints. In The Twelfth International Conference on Learning Representations, 2024

**Experimental Designs Or Analyses:**

The experimental designs and analyses in the paper appear to be sound and valid.

**Methods And Evaluation Criteria:**

Yes, the proposed methods and evaluation criteria are appropriate for the problem. The Diffusion-$\chi^n$PO method is specifically designed to address the challenge of aligning diffusion models with human preferences in text-to-image generation, and the use of $f_{\chi^n}$-regularization is a logical approach to balance optimization between preferred and non-preferred samples. The evaluation criteria, including metrics such as HPSV2, PickScore, CLIP, and Image Reward, are relevant and widely used in the field, providing a comprehensive assessment of the model's alignment performance and visual quality.

**Other Comments Or Suggestions:**

1. In line 246-249, the claim that a smaller gradient ratio can lead to misalignment when it falls below 1 is intriguing. It would be helpful if the authors could provide either a reference to relevant literature or an intuitive explanation for this phenomenon. From my understanding, a faster decrease in the probabilities of less preferred images seems reasonable, but I'm curious about the potential misalignment it might cause.

2. Have the authors considered adding a SFT loss on top of DPO? Figure 3 indicates that the choice of n significantly impacts model performance. The authors state that (line 258-261) n encourages fine-tuned diffusion models to prioritize human-preferred images while reducing the penalization of less preferred behaviors. However, I believe that directly adding an SFT loss could achieve a similar effect, and this technique has been attempted in several papers [1, 2].

3. How should we choose an appropriate n? Since n is crucial, how should we tune this parameter? While the authors propose  $\chi^n$PO as a solution, I suggest a comprehensive comparison with related technical solutions ([1, 2]) to provide a more complete picture.

[1] Noise Contrastive Alignment of Language Models with Explicit Rewards. In NeurIPS 2024.
[2] Sail into the Headwind: Alignment via Robust Rewards and Dynamic Labels against Reward Hacking. In ICLR 2025.

**Other Strengths And Weaknesses:**

see Comments.

**Questions For Authors:**

see Comments.

**Relation To Broader Scientific Literature:**

N/A

**Theoretical Claims:**

While I did not verify the detailed proofs, the derivations and arguments presented are clear and logically consistent.

---

> ### Author Rebuttal · Authors · 2025-04-01
>
> > **C1.  It would be helpful if the authors could provide either a reference to relevant literature or an intuitive explanation for this phenomenon.**
>
> Contrastive Nature of the DPO Loss: The occurrence of the same token in both the selected and rejected responses induces contradictory learning objectives, as the model is forced to simultaneously increase and decrease the probabilities of these tokens [1]. In the output of $\log\sigma(y_1 - y_2),$ where $y_1$ contributes relatively little to the overall result and the gradient ratio is significantly below 1, the magnitude by which the probability is increased when the selected and rejected responses share the same token is far less than the magnitude by which it is decreased, thereby introducing additional uncertainty [2].
>
> > **C2. I believe that directly adding an SFT loss could achieve a similar effect, and this technique has been attempted in several papers .**
>
> I once experimented with the DPO+SFT loss. This method adds a negative log-likelihood loss term to the DPO loss to prevent the log probability of the selected responses from decreasing. However, it does not fundamentally resolve the issue in DPO where the log(z) function causes a negative exponential amplification when lowering $Z_2$. This, in turn, results in $y_2$ dominating $y_1$ in the output of $\log \sigma(y_1-y_2)$. Due to the larger gradient associated with $y_2$ during updates, the optimization force to lower the probability of the rejected responses becomes dominant, causingthe parameter update direction to deviate from the desired optimization objective. Additionally, the misalignment issue described in Problem 1 still persists. Lamm3[1] had to mask out special formatting tokens during the loss calculation when using DPO+SFT to stabilize the training.
>
> χⁿPO uses the linking function $\phi_{\chi^n}$ to amplify $Z_1$ in selected responses and constrain $Z_1$ in rejected responses, thereby adjusting the contributions of $y_1$ and $y_2$ in the output of $\log \sigma (y_1-y_2)$. It further adjusts the update strength by increasing the positive gradients and decreasing the negative gradients, ensuring that the parameter update direction aligns more closely with the desired optimization objective. Even when both positive and negative gradients exist for the same token, the overall update direction still leans towards the positive.
>
> >**C3. How should we choose an appropriate n?**
>
> The parameter n is determined based on the preference strength of the selected responses relative to the rejected responses in the dataset.
>
> >**Since n is crucial, how should we tune this parameter?**
>
> When the strength of the preferred responses is high and balanced optimization between $y_1$ and $y_2$ is desired, smaller n values (e.g., 1, 2, or 3) are used. In contrast, when the strength of the preferred responses is low, larger n values (e.g., 8, 9, 10, or greater) are adopted to amplify the contribution of the preferred term in the overall loss and effectively prevent over-adjustment of the non-preferred responses.
>
> >  **I suggest a comprehensive comparison with related technical solutionsto provide a more complete picture.**
>
> Our work focuses on image generation, while existing technical solutions primarily target large language models (LLMs). Extending these approaches to diffusion models requires a significant amount of effort, so we are unable to provide a complete comparison in this discussion.
>
>
>  >**they should provide more analysis on the advantages of $\chi^n$PO compared to common divergences,**
>
>
> We introduced the commonly used JS divergence, Forward KL (FKL), and Reverse KL, and added them to Figure 1 for comparison, with an anonymous screenshot provided for reference. Additionally, in [3], experiments demonstrated that JS divergence (JSD) outperforms Reverse KL.
>  By examining the screenshots (see https://imgur.com/a/VSORILO), it can be observed that, due to the inherent properties of the logarithm, the increase in $Z_1$ and the decrease in $Z_2$ lead to inconsistent magnitudes of change in $y_1$ and $y_2$. Therefore, when converting the Reverse KL to JS divergence, the reduction in $y_2$ is greater than the decrease in $y_1$. As a result, in $\log \sigma(y_1-y_2)$ the relative contribution of $y_1$ is increased compared to the Reverse KL model.
>
> The linking function $\phi_{\chi^n}$ of $\chi^n\mathrm{PO}$ shifts the zero point to the right and employs a curve-flattening strategy in the (0,1) interval to achieve a smooth transition, thereby more significantly reducing $y_2$. In the $(1,\infty)$ interval, unlike the approach of the JS divergence, $\chi^n\mathrm{PO}$ amplifies $y_1$ by applying polynomial growth to the preferred term $Z_1$.
>
>
>
> **References**
>
> [1] The Llama 3 Herd of Models
>
> [2]Unintentional Unalignment: Likelihood Displacement in Direct Preference Optimization
>
> [3]Generalizing Alignment Paradigm of Text-to-Image Generation with Preferences through f -divergence Minimization

---

### Official Review · Reviewer_mGyc · 2025-03-11

**Overall Recommendation:** 3

**Summary:**

This paper proposes XPO, a framework to align T2I diffusion models with human preferences.  XPO introduces novel regularization techniques to smooth the training process. The authors show that XPO is more resident to conflicting samples in the training data from a theoretical perspective, and provided empirical evidences to show its effectiveness of a wide range of benchmarks.

**Claims And Evidence:**

The claim of this paper comes in two-fold. First, the author argued that XPO exhibits many theoretical advantages and provided a thorough investigation on how the proposed loss affect in the gradient field. Second, the author showed that the proposed method empirically outperform multiple baselines in the field on various benchmarks.  I find the argument and evidence sufficiently convincing.

**Essential References Not Discussed:**

N/A

**Experimental Designs Or Analyses:**

See sec "Methods And Evaluation Criteria*". Overall, I find the experiments sound and comprehensive.

**Methods And Evaluation Criteria:**

The proposed method is compressive and incorporate multiple dataset. However, the statistic significance of the results can be hard to tell at times. For example, in table 3, all win rates above 50 are bolded, including ones that are only marginally above 50%. It's hard to judge the significance of these results. The authors are suggested to conduct a thorough statistic analysis on the significance of these results. Doe they actually show that XPO is better? Or are they a statistic tie.

**Other Comments Or Suggestions:**

N/A

**Other Strengths And Weaknesses:**

N/A

**Questions For Authors:**

N/A

**Relation To Broader Scientific Literature:**

The paper clearly show empirical advantages over many SoTA baselines in the field, such as Diffusion-DPO and Diffusion-KTO. It offers valuable insights to the community, and may benefit future works on preference alignment.

**Theoretical Claims:**

I checked Section 4 and did not find major issues.  Appendix are not thoroughly checked.

---

> ### Author Rebuttal · Authors · 2025-03-31
>
> >  For example, in table 3, all win rates above 50 are bolded, including ones that are only marginally above 50%. It's hard to judge the significance of these results. The authors are suggested to conduct a thorough statistic analysis on the significance of these results. Doe they actually show that XPO is better? Or are they a statistic tie.
>
>
> We used 8,667 high-quality prompts from the [open-image-preferences-v1-binarizeds dataset](https://huggingface.co/blog/image-preferences) to generate images. We report both the reward evaluation results of the generated images and the automatic win rates of Diffusion-χ⁶PO (SD v1-5) compared to existing alignment approaches.
>
> | **Model/Score**      | HPSV2↑  | PickScore↑ | Aesthetic↑ | CLIP↑  | Image Reward↑ |
> | -------------------- | ------- | ---------- | ---------- | ------ | ------------- |
> | SD v1-5              | 24.9125 | 19.6138    | 5.7544     | 0.1654 | -0.9994       |
> | Diffusion-DPO        | 25.0342 | 19.7352    | 5.88       | 0.1641 | -0.9038       |
> | Diffusion-KTO        | 25.2431 | 19.7356    | 5.9540     | 0.1587 | -0.6633       |
> | SPIN-Diffusionsion   | 25.1551 | 19.7217    | 6.0418     | 0.1622 | -0.7940       |
> | SePPO                | 25.219  | 19.8050    | 6.062      | 0.1594 | -0.6633       |
> | Diffusion x6po(ours) | 25.2081 | 19.8789    | 5.9377     | 0.1615 | -0.6559       |
>
> | **Model/Score**        | HPSV2↑       | PickScore↑   | Aesthetic↑   | CLIP↑        | Image Reward↑ |
> | ---------------------- | ------------ | ------------ | ------------ | ------------ | ------------- |
> | vs. SD v1-5            | **74.4144%** | **73.0587%** | **65.7552%** | 44.6983%     | **77.0509%**  |
> | vs. Diffusion-DPO      | **65.9455%** | **64.0129%** | **55.8671%** | 46.8444%     | **71.3973%**  |
> | vs.  Diffusion-KTO     | 46.8444%     | **65.6282%** | 48.9097%     | **54.7710%** | **50.0404%**  |
> | vs. SPIN-Diffusionsion | **55.9305%** | **65.3283%** | 39.9677      | 49.1058      | **62.5014%**  |
> | vs. SePPO              | 48.6443%     | **58.1401%** | 36.217%      | **54.1018%** | **50.4211%**  |

---

> > ### Comment · Reviewer_mGyc · 2025-04-02
> >
> > Thanks for the response. I keep my recommendation for acceptance

---

### Official Review · Reviewer_VGbe · 2025-03-12

**Overall Recommendation:** 2

**Summary:**

The authors extend chi-square preference optimization to text-to-image tasks using diffusion models. To encompass a broader class of probability divergences, they generalize chi-square divergence to the chi-n function for positive integers n>1 and analyze the gradient of the proposed chi-n preference optimization. Finally, they evaluate the method on the HPDv2 benchmark and PartiPrompts datasets, reporting the results across various metrics.

**Claims And Evidence:**

The claim that the proposed fine-tuning approach mitigates over-optimization and enhances training efficiency is not clearly justified.

**Essential References Not Discussed:**

The related literature is discussed completely.

**Experimental Designs Or Analyses:**

I have checked the validity of all experiments.

**Methods And Evaluation Criteria:**

This paper generally follows the standard protocol for evaluating the proposed methods in terms of datasets and evaluation metrics. However, it lacks a user study, which is crucial for visual-based preference optimization.

**Other Comments Or Suggestions:**

Typos
1. q(x_{t-1,t}|x^+_0) -> q(x_{t-1,t}|x^-_0) (line 821,829,835)
2. Lemma 1 (line 663-668) has repeated statements.
3. Many spaces, periods, and sentence-initial capitalizations are missing. For example (lines 230, 234, and 325).

**Other Strengths And Weaknesses:**

**Strengths**:

This paper presents a comprehensive review of the literature and theoretical background on Chi-square preference optimization, clearly articulating the motivation behind the study—extending the constraints on preference optimization in diffusion models from KL divergence to f-divergence. Following a standard approach, the authors provide detailed derivations to support this adaptation process and further generalize it to the chi-n function. Finally, they offer an initial exploration of the gradient properties of chi-n preference optimization.

**Weaknesses**:

1. In Section 4.3 (lines 255–259), the authors state that a larger n can prevent the z value from being excessively amplified. However, Figure 1 (left) shows that increasing n causes y to grow rapidly, whereas DPO exhibits the smoothest curve.

2. The final part of Section 4.3 (lines 263–265) suggests that a larger n improves training efficiency. While this effect is somewhat observable in Figure 3, there is no comparison with other methods, making it unclear whether the proposed approach indeed enhances training efficiency.

3. The quantitative results in Table 3 indicate that the proposed method underperforms in terms of Aesthetic scores. In particular, when compared to SPIN-Diffusion, the win rate is only around 30%–35%. What factors contribute to this phenomenon?

4. Using rewards as evaluation metrics has inherent limitations, such as reward hacking [1]. Therefore, automatic win rates alone cannot serve as a definitive measure of preference. A user study is necessary to validate the reported win rate results.
[1] Gao, Leo, John Schulman, and Jacob Hilton. "Scaling laws for reward model overoptimization." In ICML 2023.

**Questions For Authors:**

Please refer to the weakness. Any clarification is welcomed.

**Relation To Broader Scientific Literature:**

This paper extends chi-square preference optimization from LLMs to diffusion models. The proposed chi-n preference optimization could benefit future research on preference optimization problems, including those in LLMs.

**Theoretical Claims:**

I have verified the correctness of the derivation of chi-n preference optimization (Appendices B, C, and D).

---

> ### Author Rebuttal · Authors · 2025-03-31
>
> > **W1 . In Section 4.3 (lines 255–259), the authors state that a larger n can prevent the z value from being excessively amplified. However, Figure 1 (left) shows that increasing n causes y to grow rapidly, whereas DPO exhibits the smoothest curve.**
>
> As the alignment process progresses, the value of the preferred component $Z_1$ gradually tends to exceed 1, while the value of the non-preferred component $Z_2$ tends to fall below 1. Moreover, compared to increasing $Z_1$ to a high value (for example, up to 2), it is easier to decrease $Z_2$ to a low value (for example, down to 0.5).
>
> The logarithmic function in DPO exhibits different behaviors in different numerical ranges. In the interval (0,1), as $Z_2$ decreases, the value of $\log(z_1)$ drops rapidly (tending toward negative infinity). In contrast, in the interval $(1,\infty)$, as $Z_1$ increases, the rate at which $\log(Z_1)$ rises is much slower than linear growth. This results in $y_2$ being significantly larger than $y_1$ in the output of $\log \sigma(y_1-y_2)$, meaning that the overall input is mainly determined by $y_2$.
> In comparison, the linking function $\phi_{\chi^n}$ of $\chi^n\mathrm{PO}$shifts the zero point to the right relative to the logarithmic function in DPO. In the interval (0,1), it adopts a curve-flattening strategy to achieve a smooth transition, thereby mitigating the negative exponential amplification issue caused by the reduction of the non-preferred term $Z_2$ in DPO. In the interval $(1,\infty)$, it amplifies $Z_1$ through polynomial-level growth, which in turn increases $y_1$.
>
> > **W2. The final part of Section 4.3 (lines 263–265) suggests that a larger n improves training efficiency. While this effect is somewhat observable in Figure 3, there is no comparison with other methods, making it unclear whether the proposed approach indeed enhances training efficiency.**
>
> The scores from each checkpoint during Diffusion-DPO training have been added to Figure 3, further confirming that a larger n can indeed improve training efficiency. Additionally, an anonymized screenshot has been uploaded below: [https://imgur.com/a/c1fcqGD](https://imgur.com/a/c1fcqGD)
>
> >**W3. The quantitative results in Table 3 indicate that the proposed method underperforms in terms of Aesthetic scores. In particular, when compared to SPIN-Diffusion, the win rate is only around 30%–35%. What factors contribute to this phenomenon?**
>
> In the[Pickapic](https://huggingface.co/datasets/yuvalkirstain/pickapic_v1)  dataset, the images labeled as preferred in the test set have an aesthetic score accuracy of only 56.8%, which results in a lack of significant differentiation in aesthetic preference between the selected and rejected images. This, in turn, makes it difficult for DPO loss–based contrastive training to significantly improve the aesthetic score.
> Meanwhile, SPIN-Diffusion utilizes a self-play fine-tuning strategy, further amplifying its advantage in aesthetic scores over the $\chi^n$PO method through iterative refinement.
>
> >**W4. A user study is necessary to validate the reported win rate results.**
>
> We conducted a comparison of the generated results on the Amazon Mechanical Turk platform. Annotators were asked to compare images in two aspects: Q1 Prompt Alignment (“Which image better fits the text description?”) and Q2 Visual Appeal (ignoring the prompt, “Which image is more visually appealing?”). The images compared were generated for the same prompt using the Diffusion $\chi^n$PO model and Diffusion-DPO. Human Evaluation Win Rate
>
> | Dataset                              | **vs.Model/Score** | Visual Attractiveness, Excluding Prompts↑ | Prompt Alignment↑ |
> | ------------------------------------ | ------------------ | ----------------------------------------- | ----------------- |
> | HPS                                  | vs. Diffusion-DPO  | 64.1%                                     | **53.1%**         |
> | open image-preferences-v1-binarizeds | vs. Diffusion-DPO  | 64.8%                                     | 49.1%             |

---

### Official Review · Reviewer_P4rN · 2025-03-13

**Overall Recommendation:** 2

**Summary:**

This paper introduces Diffusion-$\chi^n$PO, a method to align text-to-image (T2I) diffusion models with human preferences. The core idea is based on generalized preference optimization with $\chi^2$ divergence, where the author generalizes to $\chi^n$ to control the regularization for over-optimization issues reside in original preference optimization, which relies on KL divergence. Experimental results demonstrate that Diffusion-$\chi^n$PO improves alignment between textual prompts and generated images compared to existing methods such as Diffusion-DPO, SPO, and SePPO. Specifically, authors fine-tune Stable Diffusion v1.5 on the Pick-a-Pic dataset and show significant improvements in various quantitative metrics (PickScore, HPSV2, Aesthetics, CLIP, and ImageReward) as well as qualitative outputs.

**Claims And Evidence:**

The major claims of the paper are given as follows: 1) Diffusion-$\chi^n$PO achieves improved alignment with human preferences by generalizing $\chi$PO to a broader regularization family, (2) mitigates reward over-optimization typically seen in KL-regularized methods, and (3) results in better performance across multiple quantitative metrics on standard evaluation datasets. The evidence provided to support these claims includes both theoretical analysis and empirical results. Theoretical insights are offered through analyses of gradient ratios under different regularization link functions, showing how $\chi^n$PO can balance the optimization of preferred and non-preferred samples. Empirically, the authors demonstrate improvements on established benchmarks, showing consistent gains in reward scores and alignment metrics over several baselines.
However, while the improvements are convincing, the claim that Diffusion$\chi^n$PO mitigates reward over-optimization issue has not been thoroughly investigated. Also, the evaluation could be strengthened with additional results such as involving different models (e.g., recent SOTA T2I diffusion models as SD v1.5 is kind of outdated model).

**Essential References Not Discussed:**

I do not have any concerns on the references.

**Experimental Designs Or Analyses:**

The experimental setup largely follows established protocols in the field. The use of Stable Diffusion v1.5 fine-tuned on Pick-a-Pic v2 provides a reasonable testbed for preference alignment. The authors compare against strong baselines (Diffusion-DPO, SPO, SePPO, etc.) and use fair training setups by adhering to baseline hyperparameters where appropriate. Metrics are evaluated on standard datasets (HPDv2 and PartiPrompts) across different image styles, and results are reported in a comprehensive manner. However, while the experiments are convincing, some concerns remain about generalizability. The experiments primarily focus on Pick-a-Pic v2, which may not capture broader preference diversity. Also, the base model SDv1.5 is outdated, and there are numerous T2I models that naively outperform. Providing additional empirical results on different dataset (e.g., I recall there is an open-source high-quality preference dataset in [here](https://huggingface.co/blog/image-preferences)) and applying to SOTA diffusion models (e.g., SD3, Flux, etc.) would further strengthen the paper. Also, human evaluations or tests on out-of-distribution datasets would improve confidence in the generality of the claims.

**Methods And Evaluation Criteria:**

The proposed Diffusion-$\chi^n$PO method appears well-motivated and appropriate for the stated problem. The authors adapt preference optimization frameworks with trust-region regularization, which was vastly studied in language model literatures. Specifically, the usage of $\chi^2$ divergence for language model was introduced in [1], and the author adapt and generalize into the case of diffusion models.

They perform experiments on the widely used Pick-a-Pic v2 dataset and evaluate using standard metrics such as PickScore, HPSV2, CLIP score, Aesthetics, and ImageReward, which are generally accepted in the community for evaluating alignment and generation quality. However, the reliance on reward model scores for evaluation, without significant human evaluation studies, leaves some open questions regarding real-world user preferences and robustness. Nonetheless, the methodological choices are sound and aligned with recent trends in the evaluation of diffusion model alignment.

[1] Huang, Audrey, et al. "Correcting the mythos of kl-regularization: Direct alignment without overoptimization via chi-squared preference optimization." arXiv preprint arXiv:2407.13399 (2024).

**Other Comments Or Suggestions:**

N/A

**Other Strengths And Weaknesses:**

A key strength of the paper is its novel extension of preference optimization frameworks to diffusion models, providing a theoretically grounded and empirically validated method that addresses limitations of KL regularization. The proposed $\chi^n$PO framework demonstrates originality in its mathematical generalization and offers practical insights into gradient field behavior during alignment. The writing is clear, and the methodological exposition is accessible for a technical audience.

One of the main weakness lies in the limited empirical evaluation scope. While the results on Pick-a-Pic v2 and HPDv2 datasets are popular, additional testing on broader datasets or with human evaluation would improve the significance of the findings. Furthermore, the reliance on reward model scores—known to have alignment issues themselves—raises questions about the robustness of the claims regarding improved human preference alignment. Also, the proposed evaluation does not necessarily validate the claim that $\chi^n$PO is more robust towards the reward over-optimization issues.

**Questions For Authors:**

1. Could the author elaborate how the proposed $\chi^n$PO regularizes the over-optimization problem? How does the choice of $n$ affects the over-optimization? Is there any trade-off when searching for the best hyperparameter?
2. Does the author have tried the proposed method to different diffusion models? Specifically, does the trend consistent to when dealing with more sophisticated diffusion models (e.g., SD3, Flux)?
3. Could the author provide human evaluation results?

**Relation To Broader Scientific Literature:**

The proposed paper has link to divergence-based regularization, which is one of a common method in different topics of machine learning, such as reinforcement learning or variational inference.

**Theoretical Claims:**

The paper provides a theoretical intuition behind the $\chi^n$PO objective and its corresponding link function. The authors derive the gradient ratios for different values of $n$ and compare them with those from KL and $\chi^2$ regularizations, arguing for the advantages of their approach in balancing preference optimization. I have carefully checked the derivations presented in the main paper, particularly the formulations in Equations (6)-(19) as well as the detailed derivations mentioned in the supplementary material (Supp B-E), and they appear mathematically consistent. The logic connecting the χn-divergence to the behavior of the gradient fields is plausible and in line with existing work on divergence-based regularization. Assuming correctness there, the theoretical contributions are valid and offer a meaningful extension to prior work on diffusion preference optimization.

---

> ### Author Rebuttal · Authors · 2025-04-01
>
> >**W1. additional testing on broader datasets or with human evaluation would improve the significance of the findings.**
>
> We used 8,667 high-quality prompts from the [open-image-preferences-v1-binarizeds dataset](https://huggingface.co/blog/image-preferences) to generate images.  We report both the reward evaluation results of the generated images。 For details, please refer to the response to Reviewer mGyc.
>
> >**W2. the reliance on reward model scores—known to have alignment issues themselves—raises questions about the robustness of the claims regarding improved human preference alignment.**
>
> We conducted a human evaluation of the generated images; please refer to Q3 for details.
>
> >**W3 the proposed evaluation does not necessarily validate the claim that $\chi^n$PO is more robust towards the reward over-optimization issues.**
>
> $\chi$ PO has been validated for its effectiveness on large-scale language models. In contrast, $\chi^n$PO addresses the negative exponential amplification issue of $\log(z)$ in the (0,1) interval in DPO by modifying only the linking function of $\chi$ PO, and it increases the proportion of $y_1$ in the output of $\log \sigma(y_1-y_2)$, thereby aligning the parameter update direction more closely with the desired optimization objective. Experimental results further confirm the validity of this strategy.
>
>
> >**Q1.  Could the author elaborate how the proposed $\chi^n$PO regularizes the over-optimization problem?**
>
> The link function $\phi_{\chi^n}$ of $\chi^n\mathrm{PO}$ shifts the zero point relative to the logarithmic function used in DPO. It employs a curve-flattening strategy over the interval (0,1) to ensure a smooth transition, thereby mitigating the negative exponential amplification issue in DPO that arises when the value of the non-preferred term $Z_2$ decreases. In the interval $(1,\infty)$, the function amplifies the preferred term $Z_1$ via polynomial growth. This approach allows for fine control over the proportion of $y_1$ and $y_2$ in the output of $\log \sigma(y_1-y_2)$, adjusting the update magnitude by increasing the positive gradient and reducing the negative gradient. Consequently, it achieves a more balanced optimization between the preferred and non-preferred responses, ensuring that the parameter update direction better aligns with the intended optimization objectives and improves data utilization efficiency.
>
> >**How does the choice of $n$ affects the over-optimization?**
>
> By increasing the value of n, the proportion of $y_1$ in the output of $\log\sigma(y_1-y_2)$ can be raised, while positive gradients are amplified and negative gradients reduced. This adjustment in update strength ensures that the parameter update direction aligns more closely with the desired optimization objective .
>
> >**Is there any trade-off when searching for the best hyperparameter?**
>
>  As $n$ increases, the amplification effect of the regularization term $f_{\chi^n}$ on the differences between the selected response model and the initial model becomes increasingly pronounced. To mitigate this effect, we reduce the hyperparameter $\beta$  from 2000 to 1000, thereby decreasing the weight of the regularization term.
>
> >**Q2 Does the author have tried the proposed method to different diffusion models? Specifically, does the trend consistent to when dealing with more sophisticated diffusion models (e.g., SD3, Flux)?**
>
> We have not yet validated this on more complex diffusion models, and due to limited computational resources, we were unable to test this conclusion on larger diffusion models during our discussion.
> However, $\chi$PO has already demonstrated its effectiveness on large-scale language models. In contrast, $\chi^n$PO only modifies the linking function of  $\chi$PO to adjust the preference update strength, and the experimental results on the diffusion model sd-15 further confirm the validity of this strategy. In theory, $\chi^n$PO should be equally applicable to more complex diffusion models.
>
> > **Q3 Could the author provide human evaluation results?**
>
> We conducted a comparison of the generated results on the Amazon Mechanical Turk platform. Annotators were asked to compare images in two aspects: Q1 Prompt Alignment (“Which image better fits the text description?”) and Q2 Visual Appeal (ignoring the prompt, “Which image is more visually appealing?”). The images compared were generated for the same prompt using the Diffusion $\chi^n$PO model and Diffusion-DPO. Human Evaluation Win Rate
>
> | Dataset                              | **vs.Model/Score** | Visual Attractiveness, Excluding Prompts↑ | Prompt Alignment↑ |
> | ------------------------------------ | ------------------ | ----------------------------------------- | ----------------- |
> | HPS                                  | vs. Diffusion-DPO  | 64.1%                                     | **53.1%**         |
> | open image-preferences-v1-binarizeds | vs. Diffusion-DPO  | 64.8%                                     | 49.1%             |

---

### Decision · Program_Chairs · 2025-05-01

**Decision:**

Reject

**Comment:**

This paper introduces Diffusion-xPO, a method that generalizes preference optimization for text-to-image diffusion models using x^n divergence to improve alignment and mitigate over-optimization. Reviewers consistently agree with the method's novelty and theoretical grounding. Experiments demonstrated improvements over baselines on standard metrics using SD1.5.

However, major concerns were raised by several reviewers on the limited empirical studies, particularly the experimental setting of the outdated SDv1.5 base model. This was clearly mentioned by reviewers but not addressed in discussion. Additionally, relying only on automatic scores was also a concern. The authors partly addressed this later with human evaluations, showing people preferred their method to Diffusion-DPO. Questions also remained about validating the claimed mitigation of reward over-optimization.